# HO-Cap: A Capture System and Dataset for 3D Reconstruction and Pose Tracking of Hand-Object Interaction

**Jikai Wang**[1]    **Qifan Zhang**[1]    **Yu-Wei Chao**[2]    **Bowen Wen**[2]    **Xiaohu Guo**[1]    **Yu Xiang**[1]

[1]The University of Texas at Dallas
{jikai.wang, qifan.zhang, xguo, yu.xiang}@utdallas.edu

[2]NVIDIA
{ychao, bowenw}@nvidia.com

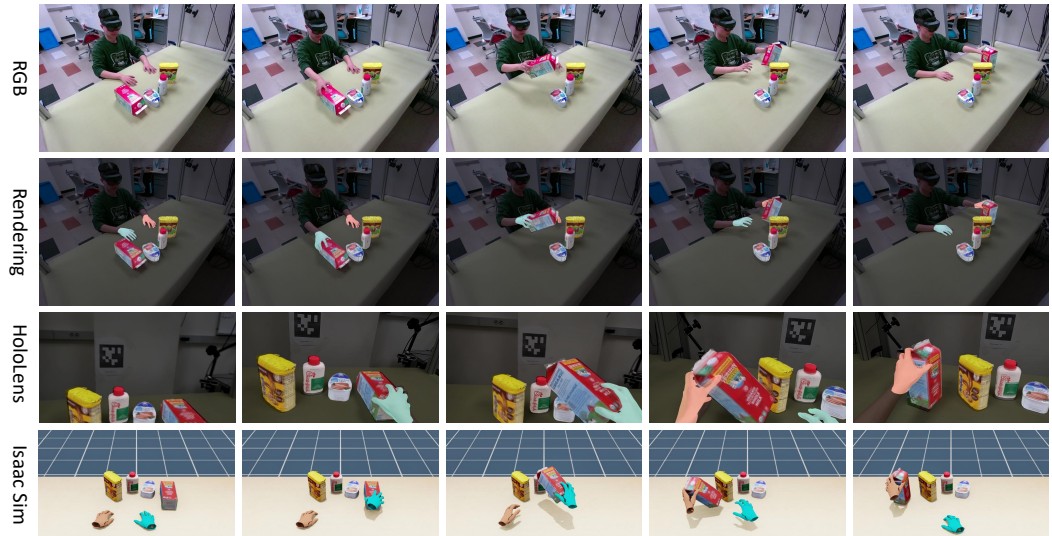

Figure 1: Sample capture from HO-Cap. From top to bottom: RGB frames, overlaid renderings of 3D shape and pose for hands and objects in third-person and egocentric views, and renderings of the capture within a virtualized scene in the NVIDIA Isaac Sim simulator.

## Abstract

We introduce a data capture system and a new dataset, **HO-Cap**, for 3D reconstruction and pose tracking of hands and objects in videos. The system leverages multiple RGB-D cameras and a HoloLens headset for data collection, avoiding the use of expensive 3D scanners or motion capture systems. We propose a semi-automatic method for annotating the shape and pose of hands and objects in the collected videos, significantly reducing the annotation time and cost compared to manual labeling. With this system, we captured a video dataset of humans performing various single- and dual-hand manipulation tasks, including simple pick-and-place actions, handovers between hands, and using objects according to their affordance. This dataset can serve as human demonstrations for research in embodied AI and robot manipulation. Our capture setup and annotation framework will be made available to the community for reconstructing 3D shapes of objects and human hands, as well as tracking their poses in videos.[1]

---

[1]Data, code, and videos for the project are available at `https://irvlutd.github.io/HOCap`.

39th Conference on Neural Information Processing Systems (NeurIPS 2025) Track on Datasets and Benchmarks.

# 1  Introduction

Hand-object interaction has been a key area of research with broad applications in human-computer interaction, VR/AR, and robot learning from human demonstration. Specific problems, such as hand detection [36, 58], hand pose estimation [18, 19], hand shape reconstruction [39, 49, 42], hand-object detection [45, 11], object pose estimation [53, 52, 30, 7], and object shape reconstruction [51, 56, 9], are actively studied within the community. To facilitate research and benchmarking of these problems, several datasets related to hands and objects have been introduced [8, 28, 55, 33, 24, 14, 10, 4]. Most datasets consist of videos of users manipulating objects in front of cameras, with ground-truth annotations such as hand poses and object poses obtained through various methods.

An easy way to obtain pose annotations is to use motion capture (mocap) systems [5, 14, 4]. However, mocap systems are not only expensive, but also require the use of artificial markers on hands and objects during data capture. Alternatively, several datasets, such as DexYCB [8], OakInk [55] and HOI4D [33], rely on manual labeling. Given the large number of video frames, human annotation is highly time-consuming. Recently, a few systems have been proposed to automatically or semi-automatically generate pose annotations for hands and objects [21, 28]. However, these systems are not scalable to a wide variety of objects or hand-object interactions. For example, HO-3D [21] is limited to objects with known 3D models and cannot handle unseen objects. Similarly, H2O [28] requires training an object tracker for each captured object, making it difficult to scale up.

In this work, we introduce a new capture system for hand-object interaction. The system utilizes eight calibrated RGB-D cameras and a HoloLens headset [1] to provide both third-person and egocentric views. We propose a semi-automatic annotation method that can accurately obtain 3D shape and pose annotations for hands and objects in videos. Unlike previous systems, our method does not rely on motion capture markers or expensive 3D scanners for reconstructing 3D object models, and it requires no domain-specific training. These properties make our capture system scalable and easily deployable for hand-object interaction, surpassing existing approaches [21, 28].

Specifically, we leverage recent large pre-trained vision models, including MediaPipe [35] for hand detection and pose estimation, BundleSDF [50, 51] for 3D object reconstruction and pose estimation, FoundationPose [52] for object pose tracking, and SAM2 [43] for object segmentation and tracking. To address noise and errors, we utilize multi-view consistency across 8 RGB-D cameras. Additionally, we propose a Signed Distance Field (SDF)-based optimization method to refine both hand and object poses in 3D space. Consequently, our annotation pipeline can automatically process captured RGB-D videos to generate 3D shapes and poses for hands and objects. The only human annotation required is to manually select two points for each object in the first frame to generate an initial segmentation mask using SAM2 [43], and label the object name to register it in our database.

Using our capture system and annotation method, we created a new dataset called HO-Cap for hand-object interaction research. The dataset includes human demonstration videos of uni- and bi-manual interactions with objects, covering three interaction types: affordance-driven object use, pick-and-place, and handovers. It contains 64 videos with 656K RGB-D frames, captured from 9 subjects interacting with 64 objects. Ground-truth annotations of 3D shape and pose for hands and objects are provided for every frame. Fig. 1 shows some examples, where we also render the annotations in the NVIDIA Isaac Sim simulator. Our dataset serves as a valuable benchmark for various hand-object recognition tasks. Specifically, we present baseline results for CAD-based object detection, open-vocabulary object detection, hand pose estimation, and unseen object pose estimation. The dataset can be used for training models for hand-object interaction or be used to test zero-shot capabilities of models trained on external data, enabling evaluation of large models for hand and object recognition. Additionally, the hand and object trajectories in the dataset can be used as human demonstrations for research in embodied AI and robot manipulation.

The contributions of this work are as follows:

- We introduce a data capture system and a semi-automatic annotation method for obtaining 3D shapes and poses of hands and objects from multi-view RGB-D videos.

- We release a new dataset for hand-object interaction, focusing on humans performing tasks with objects. It covers diverse grasping and multi-object rearrangement tasks, which are novel and valuable for the imitation learning community.

- We provide a benchmark with baseline results for object detection, hand pose estimation, and object pose estimation, which can benefit future research using our dataset.

Table 1: Comparison of our HO-Cap with recent hand-object interaction datasets.

| dataset | year | modality | #seq. | #frames | #subj. | #obj. | #views | real image | marker-less | bi-manual | object reconst. | task | label |
|---|---|---|---|---|---|---|---|---|---|---|---|---|---|
| FPHA [17] | 2018 | RGB-D | 1,175 | 105K | 6 | 4 | ego | ✓ | ✗ | ✗ | ✗ | multi-task | mocap |
| Obman [22] | 2019 | RGB-D | – | 154K | 20 | 3K | 1 | ✗ | ✓ | ✗ | ✗ | grasping | synthetic |
| HO-3D [21] | 2020 | RGB-D | 27 | 78K | 10 | 10 | 1-5 | ✓ | ✓ | ✗ | ✗ | grasping & manipulation | automatic |
| ContactPose [5] | 2020 | RGB-D | 2,303 | 2,991K | 50 | 25 | 3 | ✓ | ✗ | ✓ | ✗ | grasping & manipulation | mocap & thermal |
| GRAB [48] | 2020 | mesh | 1,335 | 1,624K | 10 | 51 | – | ✗ | ✗ | ✓ | ✗ | grasping | mocap |
| DexYCB [8] | 2021 | RGB-D | 1,000 | 582K | 10 | 20 | 8 | ✓ | ✓ | ✗ | ✗ | grasping & handover | manual |
| H2O [28] | 2021 | RGB-D | – | 571K | 4 | 8 | 4+ego | ✓ | ✓ | ✓ | ✓ | multi-task | semi-auto |
| OakInk [55] | 2022 | RGB-D | 792 | 230K | 12 | 100 | 4 | ✓ | ✗ | ✗ | ✓ | multi-task | manual |
| HOI4D [33] | 2022 | RGB-D | 4,000 | 2,400K | 4 | 800 | ego | ✓ | ✓ | ✗ | ✓ | multi-task | manual |
| AffordPose [24] | 2023 | mesh | – | – | – | 641 | – | ✗ | – | ✗ | ✗ | multi-task | synthetic |
| SHOWMe [47] | 2023 | RGB-D | 96 | 87K | 15 | 42 | 1 | ✓ | ✓ | ✗ | ✓ | grasping | semi-auto |
| ARCTIC [14] | 2023 | RGB | 399 | 2,100K | 10 | 11 | 8+ego | ✓ | ✗ | ✓ | ✓ | bimanual manipulation | mocap |
| OakInk2 [57] | 2024 | RGB | 627 | 4.01M | 9 | 75 | 3+ego | ✓ | ✗ | ✓ | ✓ | multi-task | mocap |
| HOGraspNet [10] | 2024 | RGB-D | – | 1.5M | 99 | 30 | 4 | ✓ | ✗ | ✗ | ✗ | grasping | mocap & semi-auto |
| HOT3D [4] | 2024 | RGB-Mono | 424 | 3.4M | 19 | 33 | 2+ego | ✓ | ✗ | ✓ | ✓ | multi-task | mocap |
| Ours | 2024 | RGB-D | 64 | 656K | 9 | 64 | 8+ego | ✓ | ✓ | ✓ | ✓ | multi-task | semi-auto |

# 2 Related Work

In recent years, a number of hand-object interaction datasets have been introduced. Representative datasets are summarized in Table 1, compared to ours.

**Mocap vs. Natural capture.** A straightforward way to obtain hand and object pose is to use mocap systems. By attaching reflective markers to the hands and objects, a mocap system can track these markers to obtain the hand pose and the object pose (e.g., FPHA [17], ContactPose [5], ARCTIC [14], OakInk2 [57], and HOT3D [4]). However, mocap systems are costly, require calibration between mocap markers and image cameras, and can introduce artifacts. In contrast, our multi-camera setup captures markerless data, eliminating the need for such calibration.

**Manual labeling vs. Automatic labeling.** Large-scale datasets such as DexYCB [8], OakInk [55], and HOI4D [33] rely on manual labeling, which, though accurate, is labor-intensive. In contrast, some datasets use automated labeling (e.g., HO-3D [21]) or semi-automated labeling (e.g., H2O [28] and SHOWMe [47]). Fully automated methods often introduce annotation errors, while semi-automatic approaches integrate manual error correction [28] or initialize tracking processes manually [47]. Recently, HANDAL [20] proposed a semi-automatic pipeline for 6D object pose annotation but includes only limited dynamic human–object interactions and lacks hand annotations. HOGraspNet [10] introduced an automatic pipeline for hand pose annotation but still relies on a motion-capture system for object tracking.

In our work, we introduce a semi-automatic pipeline for annotating 3D shapes and poses of both hands and objects in videos. The only required human input is selecting two points on the object in the initial frame as prompts for SAM2 [43] segmentation. The method then automatically annotates subsequent frames. As highlighted in Table 1, our dataset contains markerless unimanual and bimanual videos, with the capability for 3D shape reconstruction of novel objects. The most similar work to ours is H2O [28]. However, unlike H2O, our approach requires no domain-specific training for object pose trackers. Instead, we leverage pre-trained vision models with a multi-camera setup, making our annotation method more scalable across diverse hand-object interactions.

# 3 Data Capture Setup

Our hardware setup (Fig. 2(a)) comprises eight Intel RealSense D455 cameras and one Microsoft Azure Kinect [2], all mounted above a table to provide comprehensive RGB-D coverage of the workspace. The higher-resolution Azure Kinect is primarily used for detailed 3D object reconstruction. We calibrated the intrinsic and extrinsic parameters of all cameras using Vicalib [3], enabling the fusion of point clouds into a unified 3D coordinate frame (Fig. 2(b)–(c)). To capture additional egocentric data, users wear a Microsoft HoloLens AR headset [1] during data collection (Fig. 2(c)). This setup allows us to record synchronized first-person and third-person RGB-D video streams, along with the 6DoF head-pose data provided by the HoloLens.

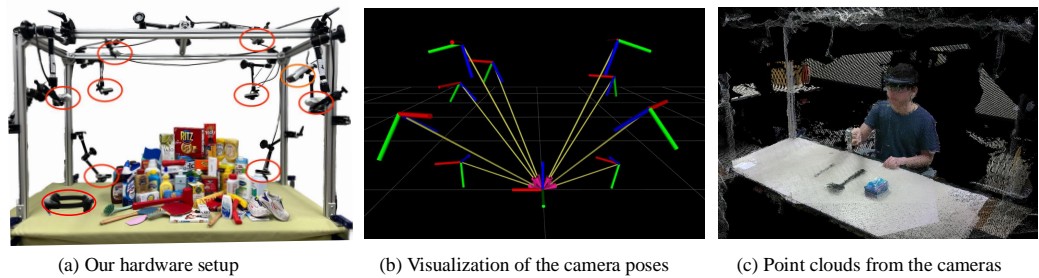

| (a) Our hardware setup | (b) Visualization of the camera poses | (c) Point clouds from the cameras |

Figure 2: Illustration of our data capture setup.

# 4 Annotation Method

Our goal is to provide 3D shapes and poses of both hands and objects in the captured videos, with the ability to handle arbitrary objects whose 3D models are not available prior to capture. To achieve this, we propose a semi-automatic annotation method based on a multi-view camera setup, eliminating the need for expensive 3D scanners or mocap systems.

## 4.1 3D Object Reconstruction

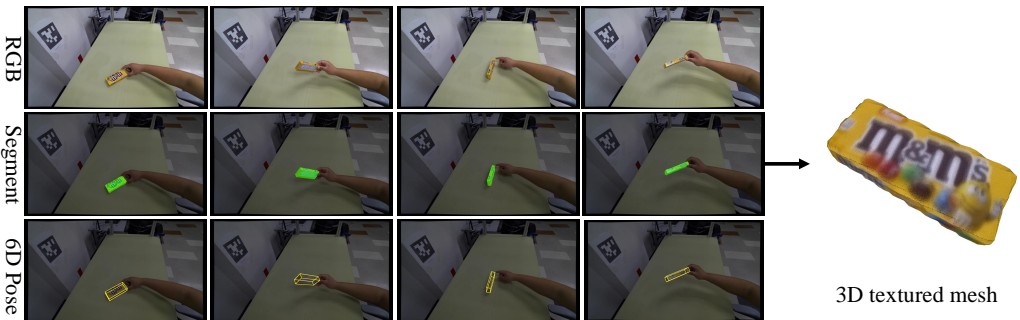

Figure 3: Illustration of our pipeline for 3D object reconstruction.

Our annotation process begins with reconstructing 3D object models. Rather than relying on existing 3D meshes (e.g., YCB Object Set [6]) or 3D scanners [47], we use BundleSDF [51], a neural reconstruction method, to reconstruct a textured 3D mesh for each object, assuming that the object is rigid. BundleSDF takes a sequence of RGB-D frames and corresponding object segmentation masks to precisely track its 6D poses and reconstruct a textured mesh. This enables us to use only an RGB-D camera for object reconstruction and obtain 3D meshes for various objects, as shown in Fig. 3. To prepare the input data, we manually move and rotate an object in front of the Azure Kinect camera, ensuring exhaustive coverage of the surfaces for high-fidelity reconstruction. For segmentation, we prompt SAM2 [43], a unified model for segmenting objects across images and videos, with two manually selected points in the initial frame to track the object mask throughout the remaining frames. Given the object masks, BundleSDF leverages feature matching based on LoFTR [46] for coarse pose initialization, followed by an online pose-graph optimization to estimate the objects' 6D poses in video frames. Simultaneously, a neural object field is trained to model object-centric geometry and appearance, refining keyframe poses to reduce tracking drift. Finally, a textured 3D mesh is extracted from the neural field using marching cubes [34] and color projection. In total, we reconstructed 64 objects in our dataset, all of which are easily available for purchase online. For visualizations of these reconstructed objects, please refer to the Appendix.

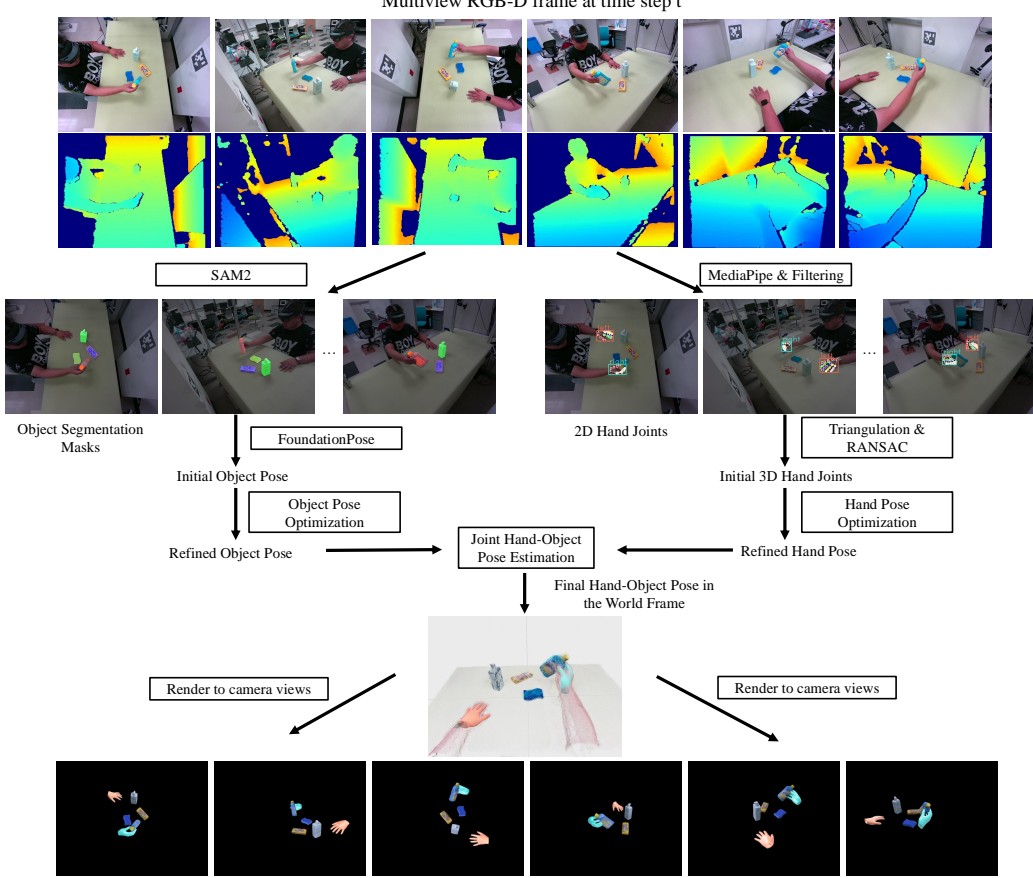

Figure 4: Our pipeline for obtaining the poses of hands and objects from multi-view RGB-D videos.

## 4.2 Object Pose Estimation

After obtaining the 3D object models, we use them to estimate object poses in videos where subjects manipulate these objects. The pipeline for hand and object pose estimation is shown in Fig. 4.

**Object Pose Initialization.** In our framework, each object's pose is represented as a homogeneous transformation $\mathbf{T} = (\mathbf{R}, \mathbf{t}) \in \mathbb{SE}(3)$, where $\mathbf{R}$ and $\mathbf{t}$ denote the object's 3D rotation and translation relative to the world frame. The world frame is defined near the center of the table (see Fig. 2(b)). Since all cameras are extrinsically calibrated, the transformations between camera frames and the world frame are known. Our pose estimation algorithm initializes each object's pose using the estimation results from FoundationPose [52], a unified foundation model for 6D object pose estimation and tracking, and then refines the pose with SDF optimization on the segmented point cloud.

Following the segmentation process described in Section 4.1, we obtain segmentation masks for each object throughout the sequence (examples shown in Fig. 4). To determine an object's initial pose at step $t$, denoted as $\mathbf{T}_t = (\mathbf{R}_t, \mathbf{t}_t)$, we use FoundationPose to track the object pose across all camera views.

Since FoundationPose operates on single-camera input and can be affected by occlusions during manipulation tasks, the tracked pose may become inaccurate, leading to tracking failures in subsequent frames for individual views. To ensure robust tracking across the video sequence, we optimize pose consistency across views by aligning tracked poses from multiple camera perspectives to reduce discrepancies and enhance accuracy.

Specifically, given the tracked pose $\mathbf{T}_{c_i}$ from each camera view $c_i$, we transform these poses from their respective camera frames into the world frame. If the distribution of pairwise distances between

all the transformed poses varies significantly, indicating that the poses are noisy, the optimization process is skipped, and the pose from the previous frame is used instead. If the distances between all transformed poses and the previous pose $\mathbf{T}_{t-1}$ are within a predefined threshold, suggesting no significant pose change, we incorporate the previous frame's pose into the RANSAC algorithm [16] to improve robustness in identifying outliers. Finally, RANSAC is applied to identify and filter out outliers based on the computed distances, resulting in a single, coherent pose for the object. The optimized initial pose $\mathbf{T}_t$ in the world frame is then projected back into each camera frame and provided as input to FoundationPose to track the object in the next frame, ensuring reliable and accurate tracking across frames.

**SDF-based Object Pose Tracking.** Although these initial poses may be effective for FoundationPose to track in the next frame, small inaccuracies and misalignments with the object point clouds can still occur. These errors can arise due to occlusions, camera noise, or drift over time. To address this, we apply a Signed Distance Field (SDF)-based algorithm to optimize the poses across the sequence. Our SDF-based pose tracking leverages the complete fused point cloud from all eight cameras, resulting in higher-quality pose annotations. At time step $t$, our goal is to optimize the initial object pose $\mathbf{T}_t$ given the previous pose $\mathbf{T}_{t-1}$, where $t = 1, 2, \ldots, T$ and $T$ is the total number of frames. We minimize the loss function

$$\mathbf{T}_t^* = \arg\min_{\mathbf{T}_t} \left( L_{\text{sdf}}(\mathbf{T}_t) + \lambda_1 L_{\text{smooth}}(\mathbf{T}_t, \mathbf{T}_{t-1}) \right), \tag{1}$$

where $\lambda_1$ is a weight to balance the two loss terms. The SDF loss function is defined as

$$L_{\text{sdf}}(\mathbf{T}_t) = \frac{1}{|\mathcal{X}_t|} \sum_{\mathbf{x} \in \mathcal{X}_t} |\text{SDF}(\mathbf{x}, \mathbf{T}_t)|^2, \tag{2}$$

where $\mathcal{X}_t$ denotes a set of 3D points of the object at time step $t$ in the world frame, which is fused from multiple camera views. The function $\text{SDF}(\mathbf{x}, \mathbf{T}_t)$ computes the signed distance of a 3D world point $\mathbf{x} \in \mathbb{R}^3$ transformed into the object frame according to $\mathbf{T}_t$. Minimizing the SDF loss function results in an object pose that aligns the transformed 3D points with the object model surface. The smoothness loss term is defined as

$$L_{\text{smooth}}(\mathbf{T}_t, \mathbf{T}_{t-1}) = \|\mathbf{q}_t - \mathbf{q}_{t-1}\|^2 + \|\mathbf{t}_t - \mathbf{t}_{t-1}\|^2, \tag{3}$$

where $\mathbf{q}_t$ and $\mathbf{q}_{t-1}$ are the quaternions of the 3D rotations, and $\mathbf{t}_t$ and $\mathbf{t}_{t-1}$ denote the 3D translations, respectively. The smoothness term prevents large jumps in the poses during optimization. By solving Eq. (1) for every object and every time step, we obtain the poses of all the objects in the video sequence. These poses provide initializations that are further refined jointly with the hands, as detailed in Sec. 4.4.

## 4.3 Hand Pose Estimation

We use the MANO model [44] to represent human hands. This parametric model enables detailed hand mesh modeling through shape and pose parameters. The shape parameters, $\boldsymbol{\beta} \in \mathbb{R}^{10}$, capture individual hand identity, reflecting variations among different subjects, while the pose parameters, $\boldsymbol{\theta} \in \mathbb{R}^{51}$, describe the dynamic positions and orientations of the hand. Before optimizing hand poses, we pre-calibrate the MANO shape parameters for each subject in our dataset (see details in the Appendix).

In our initial attempts to optimize hand poses, we applied the same approach used in Sec. 4.2 for object pose estimation, expecting similar results. However, it was challenging to obtain accurate segmentation masks using SAM2. We recognized the need to incorporate additional robust constraints into our hand pose optimization method. Specifically, we introduced 2D/3D hand keypoints as anchor points to guide the optimization process. These keypoints serve as strong priors, helping to ensure more realistic and physically plausible hand poses. At time step $t$ of an input video, our goal is to estimate the MANO hand pose $\boldsymbol{\theta}_t$ of a hand. We solve the following optimization problem to estimate the hand pose:

$$\boldsymbol{\theta}_t^* = \arg\min_{\boldsymbol{\theta}_t} \left( L_{\text{keypoint}}(\boldsymbol{\theta}_t) + \lambda_2 L_{\text{reg}}(\boldsymbol{\theta}_t) \right), \tag{4}$$

where $L_{\text{keypoint}}$ is a loss function based on the estimated 3D keypoints of the hand, $L_{\text{reg}}$ is a regularization term, and $\lambda_2$ is a weight to balance the two terms. The regularization term is simply defined as the squared L2 norm of the pose parameter: $L_{\text{reg}}(\boldsymbol{\theta}_t) = \|\boldsymbol{\theta}_t\|^2$.

**3D Keypoint Loss Function.** To generate accurate 3D keypoints for hand pose optimization, we first detect 2D hand landmarks across multiple views using MediaPipe [35]. While MediaPipe can comprehensively identify all 21 hand joints, it lacks confidence scores, making it challenging to distinguish between occluded or inaccurately estimated joints. This can introduce inaccuracies in the pose optimization process. To address these challenges, we use a RANSAC [16] based filtering approach to improve the accuracy of our 3D keypoints. For each hand joint, assume it is detected by MediaPipe on $C_{\text{valid}}$ camera views. We compute a set of candidate 3D keypoints, $\mathcal{X}_i = \{\mathbf{x}_i\}_{i=1}^{N_{\text{valid}}}$, $N_{\text{valid}} = C_{\text{valid}}(C_{\text{valid}} - 1)/2$ by triangulating pairs of valid views. A projection loss function is defined for a 3D keypoint:

$$L_{\text{proj}}(\mathbf{x}_i) = \sum_{c \in C_{\text{valid}}} \|\Pi^c(\mathbf{x}_i) - \mathbf{h}_i^c\|^2, \tag{5}$$

where $\Pi^c(\cdot)$ is the projection function for camera $c$, and $\mathbf{h}_i^c$ represents the detected 2D landmark for the 3D keypoint $\mathbf{x}_i$ in camera $c$. This loss measures the discrepancy between each candidate 3D keypoint and its corresponding 2D landmarks across views. Minimizing this projection loss refines the 3D keypoint positions, and we select the candidate with the lowest projection error for each joint. This critical step helps exclude views affected by incorrect MediaPipe detections, enhancing the reliability of the derived 3D keypoints. These optimized 3D keypoints serve as robust priors in the subsequent optimization stages.

For video frames where MediaPipe does not successfully detect hand landmarks across all camera views, we use linear interpolation between adjacent frames to estimate missing data. After filling these gaps, we further refine the hand motion trajectory using cubic spline interpolation. This approach not only smooths the spatial transitions of the hand joints but also ensures continuity in the first and second derivatives of the motion trajectory, corresponding to the hand's velocity and acceleration. By implementing this refinement, we achieve a more cohesive and realistic representation of hand motion across the entire sequence, enhancing the fluidity and naturalness of the observed actions. Using the estimated 21 3D hand joints $(\mathbf{x}_1, \ldots, \mathbf{x}_{21})$, the 3D keypoint loss function is defined as:

$$L_{\text{keypoint}}(\boldsymbol{\theta}_t) = \frac{1}{21} \sum_{i=1}^{21} \|J_i(\boldsymbol{\theta}_t) - \mathbf{x}_i\|^2, \tag{6}$$

where $J_i(\boldsymbol{\theta}_t) \in \mathbb{R}^3$ represents the $i$th 3D hand joint from the MANO model under pose $\boldsymbol{\theta}_t$. Solving Eq. (4) will find the MANO hand pose $\boldsymbol{\theta}_t$ that fits the estimated 3D keypoints.

## 4.4 Joint Hand-Object Pose Optimization

Separately solving hand and object poses can lead to unrealistic scenarios, such as intersections between hand and object meshes. To address this limitation and improve pose accuracy, we propose a joint pose optimization method that refines hand and object poses together. This approach reduces mesh intersections and enhances the physical realism of the estimated poses. For a sequence with $N_H \in \{1, 2\}$ hands and $N_O$ objects, at each time step $t$, we jointly refine the object poses $\mathcal{P}_t^O = \{\mathbf{T}_t^o\}_{o=1}^{N_O}$ and hand poses $\mathcal{P}_t^H = \{\boldsymbol{\theta}_t^h\}_{h=1}^{N_H}$. Our idea is to utilize the SDFs of objects and hands to optimize the poses as in our object pose estimation method. The loss function for this joint optimization is defined as

$$L_{\text{joint}}(\mathcal{P}_t^O, \mathcal{P}_t^H) = \frac{1}{N_O} \sum_{o=1}^{N_O} \left( \frac{1}{|\mathcal{X}_t^o|} \sum_{\mathbf{x} \in \mathcal{X}_t^o} \left| \text{SDF}_o(\mathbf{x}, \mathbf{T}_t^o) \right|^2 \right)$$
$$+ \frac{1}{N_H} \sum_{h=1}^{N_H} \left( \frac{1}{|\mathcal{X}_t^h|} \sum_{\mathbf{x} \in \mathcal{X}_t^h} \left| \text{SDF}_h(\mathbf{x}, \boldsymbol{\theta}_t^h) \right|^2 + \lambda_3 \|\boldsymbol{\theta}_t^h\|^2 \right). \tag{7}$$

where $\mathcal{X}_t^o$ and $\mathcal{X}_t^h$ denote the segmented point clouds for object $o$ and hand $h$ at time step $t$ in the world frame, while $\text{SDF}_o$ and $\text{SDF}_h$ represent the signed distance fields of the object and the hand, respectively. $\lambda_3$ is a weight to balance the regularization term for the hand pose. Additionally, the smoothness loss (Eq. (3)) is applied to ensure temporal consistency of object poses and hand global translation and rotation.

The point clouds of objects can be obtained using the depth images and the segmentation masks of the objects. To obtain point clouds for hands, given the optimized 3D hand keypoints, we can get

Table 2: Detailed dataset statistics grouped by handness (R: right hand, L: left hand, B: both hands) and task type (T1: pick-and-place, T2: handover, T3: affordance usage).

| Statistics | Handness | | | Tasks | | |
|---|---|---|---|---|---|---|
| | R | L | B | T1 | T2 | T3 |
| # Sequences | 35 | 8 | 21 | 28 | 21 | 15 |
| # Frames | 41,327 | 8,373 | 23,244 | 29,706 | 23,244 | 19,994 |

Table 3: Evaluation of annotation accuracy and physical plausibility across refinement stages. "2D/3D" indicates the mean $\pm$ standard deviation of 2D pixel error and the average 3D distance error (mm). Penetration depth (mm) and intersection volume ($cm^3$) measure physical consistency between the hand and object.

| Stage | Obj ↓ | L-Hand ↓ | R-Hand ↓ | Pen. (mm) ↓ | Vol. (cm³) ↓ |
|---|---|---|---|---|---|
| Initial | $3.67\pm3.05$ / 6.26 | $5.31\pm3.43$ / 47.38 | $5.16\pm3.73$ / 46.43 | — | — |
| SDF-Refined | $3.50\pm1.70$ / 5.01 | $\mathbf{4.58}\pm2.73$ / 14.26 | $3.83\pm2.29$ / 11.52 | $6.20\pm3.29$ | $3.88\pm2.90$ |
| Jointly-Refined | $\mathbf{3.42}\pm1.73$ / $\mathbf{3.42}$ | $\mathbf{4.58}\pm2.72$ / $\mathbf{14.25}$ | $\mathbf{3.82}\pm2.29$ / $\mathbf{11.51}$ | $\mathbf{6.01}\pm3.17$ | $\mathbf{3.45}\pm2.50$ |

the bounding box and 2D keypoints on camera images. Using these 2D keypoints along with the bounding box as input, we employ SAM2 [43] to generate high-quality hand masks. We further isolate hand points from the surrounding environment using a point-to-mesh distance threshold based on the initial hand pose. Hand and object poses are initialized from the previous stages of our annotation process, and joint optimization requires only a few refinement steps to achieve robust results. This process effectively reduces mesh intersections and enhances the realism of hand-object interactions, providing accurate and cohesive pose annotations. After estimating the poses of hands and objects in the world frame, we project their 3D shapes to the camera views and obtain 2D annotations of images as shown in Fig. 4. We also estimate the camera poses of the HoloLens in the world frame as described in Appendix B.

# 5   The HO-Cap Dataset

**Dataset Statistics.** Our HO-Cap dataset contains 64 video sequences capturing 9 participants performing three hand–object interaction tasks with 64 unique objects. The approximately 656K frames provide rich temporal information for studying dynamic interactions, where each frame is captured from 8 calibrated RealSense cameras plus a first-person-view camera from the HoloLens, facilitating 3D reconstruction and egocentric understanding. The 64 different objects, each with a textured 3D mesh model, enabling fine-grained 6D pose annotations, and a diverse set of 9 subjects ensures variability in hand shapes, grasping styles, and interaction patterns. The both single-hand and bimanual interactions, supporting tasks that require complex hand coordination. Table 2 provides handedness statistics for left/right hands and bi-manual interactions, and a breakdown of task categories. Additional details on object shape variability, hand–object pose diversity and grasp types are provided in the Appendix.

**Annotation Quality.** Using our annotation pipeline, we generated 3D shapes and world-space poses for both hands and objects. To assess annotation accuracy, we randomly selected 800 images across eight RealSense camera views and manually annotated the visible hand joints and object keypoints. The object keypoints were chosen from predefined mesh vertices that are easily identifiable. We evaluated the annotation quality using two error metrics: (1) the Euclidean distance between the 2D projections of our 3D annotations and the corresponding human-labeled 2D points, and (2) the Euclidean distance between our 3D annotations and the 3D points triangulated from the human-labeled 2D keypoints. To further evaluate physical plausibility, we measured two interaction-specific metrics introduced in [23]: penetration depth and intersection volume. Table 3 summarizes the results. The final annotation error is within 5 pixels in 2D and exhibits low 3D error for both hands and objects, demonstrating the reliability of our method. Moreover, the error consistently decreases across the refinement stages, confirming the effectiveness of our optimization strategy and the improved physical consistency between hand and object.

Table 4: Evaluation of hand pose estimation. The numbers in parentheses denote the thresholds used for PCK, and the unit of MPJPE is millimeters (mm).

| Method | PCK(0.05) ↑ | PCK(0.1) ↑ | PCK(0.15) ↑ | PCK(0.2) ↑ | MPJPE (mm) ↓ |
|---|---|---|---|---|---|
| A2J-Transformer [25] | 12.1 | 26.8 | 39.4 | 50.5 | 78.7 |
| InterWild [37] | 51.7 | 60.9 | 70.0 | 78.6 | 57.6 |
| HaMeR [42] | 43.7 | 79.2 | 88.5 | 91.4 | 28.9 |

Table 5: Evaluation of object detection. Results are reported as mean Average Precision (AP) under different IoU thresholds and object scales. Marker * denotes models trained on our dataset.

| Method | AP | $AP_{50}$ | $AP_{75}$ | $AP_S$ | $AP_M$ | $AP_L$ |
|---|---|---|---|---|---|---|
| CNOS [40] | 25.3 | 27.9 | 24.8 | 1.6 | 27.6 | 24.9 |
| GroundingDINO [32] | 17.0 | 27.6 | 21.5 | 1.4 | 24.3 | 7.5 |
| YOLO11* [26] | 71.4 | 85.9 | 78.7 | 20.7 | 75.2 | 72.6 |
| RT-DETR* [59] | 75.9 | 90.0 | 83.4 | 21.1 | 79.8 | 84.8 |

Table 6: Evaluation of object pose estimation for novel objects. Results are reported as the Area Under the Curve (AUC, %) of the ADD and ADD-S metrics on all 64 objects in our dataset.

| Method | ADD (%) | ADD-S (%) |
|---|---|---|
| MegaPose [29] | 67.1 | 83.0 |
| FoundationPose [52] | 89.3 | 95.7 |

## 6 Baseline Experiments

**Hand Pose Estimation.** First, our dataset supports hand pose estimation by providing 2D and 3D annotations for 21 hand joints. In this experiment, we evaluated three recent hand pose estimation models: A2J-Transformer [25], InterWild [37] and HaMeR [42] which are trained on external data, using ground truth hand bounding boxes as inputs. The evaluation results are presented in Table 4. We used the PCK (Percentage of Correct Keypoints) metric for 2D hand pose estimation and the MPJPE (Mean Per-Joint Position Error) metric for 3D hand pose estimation. A2J-Transformer extends the depth-based A2J [54] to RGB input and incorporates a transformer architecture to capture non-local information. This model was trained on the InterHand2.6M dataset [38]. InterWild [37] aligns motion-capture and in-the-wild hand data into shared 2D and geometry-based appearance-invariant domains, enabling robust 3D recovery of interacting hands in unconstrained environments. HaMeR [42] uses a large-scale ViT backbone [13] followed by a transformer decoder to regress the parameters of the hand, and was trained on a large-scale dataset of 2.7M images from 10 hand pose datasets. In Table 4, HaMeR significantly outperforms A2J-Transformer and InterWild, likely due to its ViT backbone and large-scale training data. The MPJPE of HaMeR is 28.9mm, which is much larger than the errors reported in other datasets in [42]. This suggests our dataset presents unique challenges for hand pose estimation, especially in cases of occlusions between hands and objects.

**Object Detection.** We evaluated object detection in two scenarios: novel object detection and seen object detection. For novel object detection, the model detects objects not encountered during training. While for seen object detection, models are trained specifically on our dataset. CNOS [40] and GroundingDINO [32] served as baselines for novel object detection, while YOLO11 [26] and RT-DETR [59] were trained for seen object detection on our dataset.

*Novel Object Detection Baselines:* CNOS [40] is a CAD-based approach that generates object templates by rendering CAD models and uses SAM [27] and DINOv2 CLS tokens [41] to classify proposals based on template similarity. GroundingDINO [32], a vision-language model, was tested with concatenated object names from our dataset as text prompts to detect objects.

*Seen Object Detection Baselines:* We trained YOLO11 and RT-DETR on our train/val split, which includes all subjects, views, and objects, and evaluated them on a test split with sequences not shared with train/val split. YOLO11, the latest in the YOLO series, features improved architecture and training methods for efficient detection. RT-DETR, a transformer-based model, is designed for robust performance across diverse object scales and complex scenes.

The results, shown in Table 5 with MSCOCO [31] Average Precision (AP) metrics, reveal challenges in novel object detection, as CNOS and GroundingDINO produced many false positives due to mismatches and ambiguities. For seen object detection, YOLO11 and RT-DETR both show particularly strong performance on medium and large objects. These results highlight the utility of our dataset for training object detectors, with RT-DETR and YOLO11 excelling on objects seen during training.

**Novel Object Pose Estimation.** Traditional object pose estimation methods [53, 12] require the same objects to be used in both training and testing, preventing to generalize to new objects. Recent novel object pose estimation approaches have been developed to address this limitation by leveraging large-scale datasets and pre-trained models. Methods such as MegaPose [29] and FoundationPose [52] can estimate poses for unseen objects using only 3D models, without per-instance training. We thus evaluated MegaPose and FoundationPose on our dataset, providing ground truth 2D bounding boxes as input. Table 6 reports the AUC of ADD and ADD-S metrics [53, 12]. FoundationPose outperforms MegaPose, likely due to its powerful transformer-based architecture and extensive synthetic training. See supplemental for more visualizations.

## 7   Conclusion and Discussion

We introduced **HO-Cap**, a novel capture system and dataset for hand-object interaction research. Our system uses multiple RGB-D cameras and a HoloLens headset to capture videos from third- and first-person views. A semi-automatic annotation pipeline integrates pre-trained vision models for object reconstruction, segmentation, pose estimation, and hand joint detection, requiring no domain-specific training. We further refine pose accuracy using SDF-based optimization. HO-Cap captures diverse hand-object interactions and offers a valuable resource for research in embodied AI and robotic manipulation.

**Limitations.** Our annotation method has three key limitations. (1) BundleSDF [51] struggles with textureless or reflective objects (e.g., metal), which were excluded from the dataset. (2) MediaPipe [35] occasionally fails to detect hand joints, preventing pose estimation in affected frames. (3) Small or cylindrical objects (e.g., spatulas, hammers) are often heavily occluded when grasped, limiting visual cues and causing pose inconsistencies across views. These challenges hinder reliable multi-view tracking, leading us to discard affected videos. Addressing these issues is an important direction for future work.

## 8   Acknowledgement

This work was supported in part by the DARPA Perceptually-enabled Task Guidance (PTG) Program under contract number HR00112220005, the Sony Research Award Program, the National Science Foundation (NSF) under Grant Nos. 2346528 and 2520553, and the NVIDIA Academic Grant Program Award.

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

# Appendix

# A Details on Data Collection

## A.1 Protocol

The HO-Cap dataset was collected with the participation of nine individuals in a controlled laboratory environment. To ensure participant privacy and data anonymization, all individuals were instructed to remove any items that could reveal their identity. No facial or biometric data was captured during the collection process.

Prior to the recording sessions, each participant was guided to calibrate their personalized MANO hand shape parameters. Additionally, participants were trained to properly wear and operate the HoloLens headset used for egocentric video capture.

During the data collection, participants were asked to perform one of three general categories of hand-object manipulation tasks: (1) pick-and-place, (2) handovers between hands, and (3) object usage based on functional affordances. To encourage natural and diverse interactions, no strict constraints were imposed on task execution. Participants were free to choose the initial object poses, the pace of their actions, and the order of object interactions. The only explicit instructions were to maintain visual focus on the manipulated object and to begin and end each task with their hands resting in a neutral pose on the table. This consistency facilitates downstream segmentation, synchronization, and initialization processes.

## A.2 MANO Shape Calibration

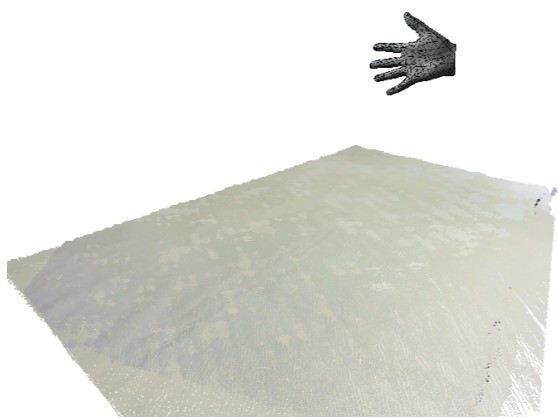

Figure 5: Visualization of the rendered MANO hand mesh used as a reference for participants during calibration.

We calibrated each subject's hand shape using a three-step process, assuming both hands share the same shape. First, we performed initial pose estimation and 3D points collection. Participants were instructed to align their right hand with a rendered neutral MANO hand mesh displayed in front of a camera (Fig. 5). During this process, 3D points around the mesh were dynamically collected and filtered based on their distance to the mesh surface, retaining only those within a predefined threshold. The hand pose was then optimized by fitting the 3D points to the signed distance field (SDF) of the MANO model, keeping the shape parameter $\beta$ fixed at zero to focus solely on pose refinement. The optimized pose and filtered 3D points were saved for the next step. Second, we iteratively optimized both the hand shape $\beta$ and pose $\theta$ using the saved pose and points. This involved alternately fixing one parameter while optimizing the other, gradually improving alignment between the hand mesh and the collected 3D point data. Finally, the optimized hand shape could be further refined by using sequences collected in the dataset. Once finalized, the calibrated hand shape was fixed throughout the following optimization process for hand pose estimation.

## A.3 Synchronization of the Camera Streams

To facilitate efficient control and data collection, all cameras are integrated into a ROS server for seamless acquisition. The 8 static RealSense cameras are connected via wired cables to the server, where they publish synchronized RGB and depth images. The HoloLens, connected via an Ethernet cable, operates in Research Mode to access and publish RGB camera streams and head tracking data over TCP/IP. All image frames and pose data are recorded into a single ROS bag file, with synchronization performed based on timestamps to ensure temporal alignment.

## B HoloLens Camera Pose Estimation

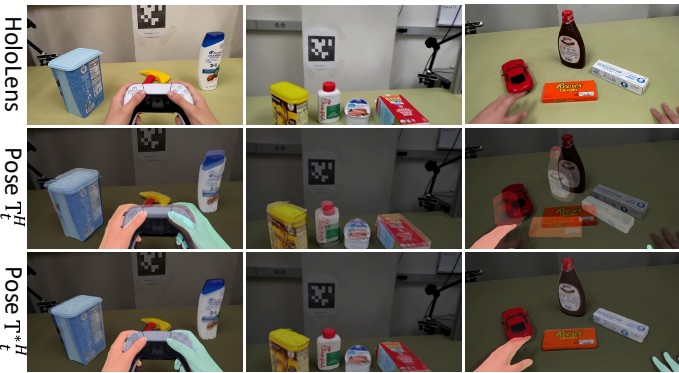

Figure 6: Comparison between the published and refined HoloLens poses

During data collection, the HoloLens head pose data often exhibited jitter and non-uniform movement speeds, making it difficult to compute precise HoloLens camera poses in real time. Therefore, we cannot directly use the pose data from HoloLens. We applied an optimization technique to refine the head poses provided by HoloLens.

At each time step $t$, we have the HoloLens camera pose $\mathbf{T}_t^H$ in the world frame published from the device. To refine this camera pose, we obtain the optimized object poses $\mathcal{P}_t^O$ as described in Sec. 4.4. By treating the group of objects as a single merged entity, we define the combined object pose as $\mathbf{T}_t^O$ in the world frame. By transforming this object pose into the HoloLens camera frame (denoted as $\mathbf{T}_t^{OH}$), and assuming proper synchronization between the HoloLens and RealSense frames, we can proceed with further refinement. Using FoundationPose [52], we optimize the object pose $\mathbf{T}_t^{OH}$ in the HoloLens camera frame using the RGB image from HoloLens, resulting in a refined camera pose $\mathbf{T}^*{}_t^H$. As shown in Fig. 6, this refinement reduces jitter and improves alignment accuracy for the projected objects in the HoloLens camera frame.

## C Properties of HO-Cap

### C.1 Annotation Details

The HO-Cap dataset provides MANO-based 3D hand pose and 6D object pose annotations, optimized within a global world frame using 8 RealSense cameras. Camera intrinsics and extrinsics for all views are included, enabling the transformation of world-frame annotations into camera-frame 3D poses. The 2D hand joint keypoints are obtained by projecting the 3D hand joints onto the image plane. Additionally, segmentation masks offer pixel-wise annotations for objects and hands, facilitating precise scene understanding. The First-Person View (FPV) data from the HoloLens headset provides egocentric perspectives crucial for human perception research and assistive AR applications.

### C.2 Visualization and Scene Simulation

The HO-Cap dataset provides textured 3D meshes for 64 objects (Fig. 7), compatible with physics simulators, enabling realistic scene reconstruction and interaction modeling. For sequence data visualization and scene replay in Isaac Sim, please refer to the supplementary video.

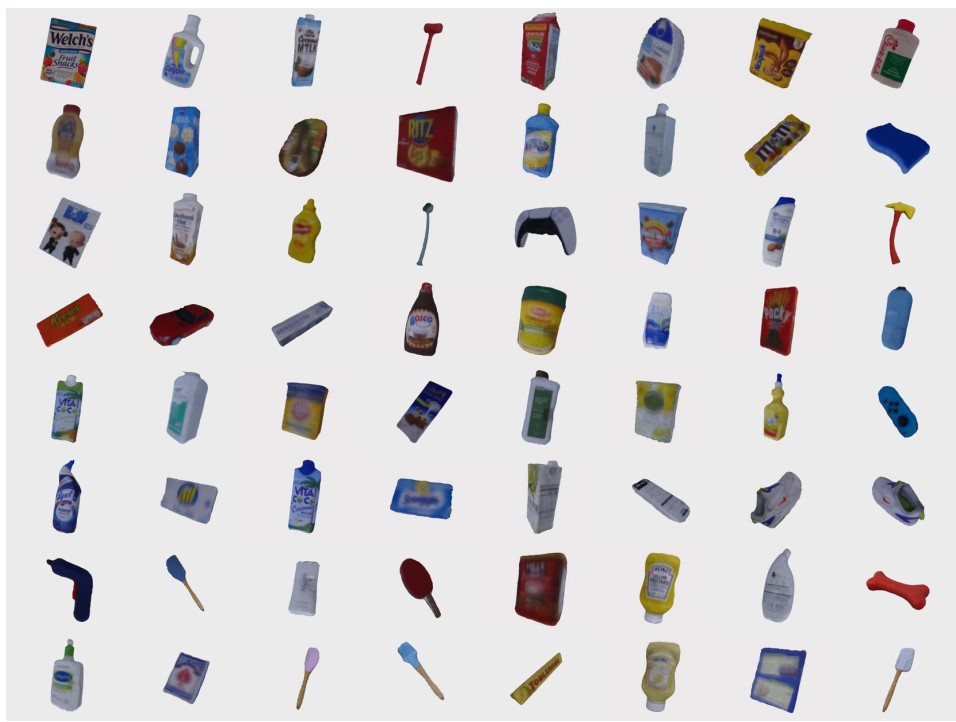

Figure 7: Reconstructed textured meshes of the 64 objects in HO-Cap

## C.3 Shape and Pose Diversity

As illustrated in Fig. 8, our dataset (1) encompasses a diverse range of hand-held object shapes and sizes. and (2) offers a broader variety of hand poses compared to HO-3D [21], capturing more natural and dynamic interactions.

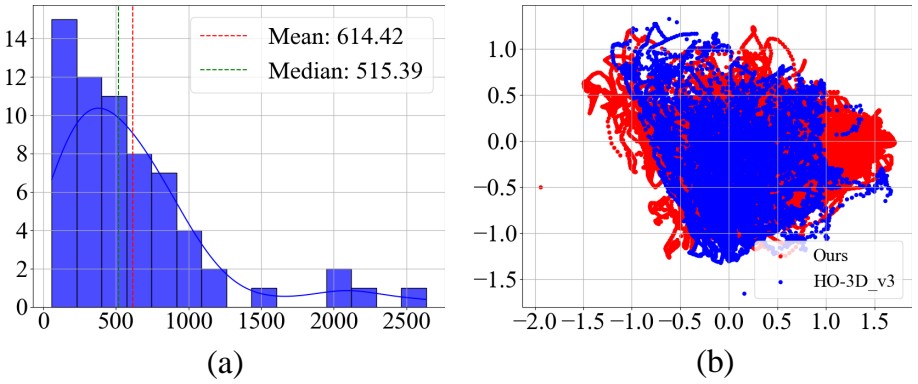

Figure 8: Distribution of (a) object volume ($mm^3$), (b) hand pose with comparison to HO-3D (first two MANO PCA coefficients).

## C.4 Grasp Types

To further validate the diversity of HO-Cap, we analyzed the dataset using the 33 grasp taxonomies defined by [15]. As shown in Table 7, HO-Cap covers 28 out of 33 grasp types—substantially more than HO-3D (15) and DexYCB (14)—demonstrating its broader coverage of human grasp behaviors. Each count indicates the number of unique sequences containing at least one instance of the corresponding grasp type. We also grouped these grasp types by task category (Table 8) to

Table 7: Grasp type distribution based on the 33 grasp taxonomies defined by [15].

| Grasp Type | #Seq | Grasp Type | #Seq | Grasp Type | #Seq |
|---|---|---|---|---|---|
| 01_Large_Diameter | 49 | 12_Precision_Disk | 18 | 23_Adduction_Grip | 2 |
| 02_Small_Diameter | 12 | 13_Precision_Sphere | 1 | 24_Tip_Pinch | 1 |
| 03_Medium_Wrap | 5 | 14_Tripod | 10 | 25_Lateral_Tripod | 0 |
| 04_Adducted_Thumb | 24 | 15_Fixed_Hook | 1 | 26_Sphere_4_Finger | 0 |
| 05_Light_Tool | 4 | 16_Lateral_Type | 6 | 27_Quadpod | 10 |
| 06_Prismatic_4_Finger | 3 | 17_Index_Finger_Extension | 19 | 28_Sphere_3_Finger | 3 |
| 07_Prismatic_3_Finger | 4 | 18_Extension_Type | 17 | 29_Stick | 8 |
| 08_Prismatic_2_Finger | 1 | 19_Distal_Type | 0 | 30_Palmar | 33 |
| 09_Palmar_Pinch | 3 | 20_Writing_Tripod | 0 | 31_Ring | 5 |
| 10_Power_Disk | 5 | 21_Tripod_Variation | 0 | 32_Ventral | 8 |
| 11_Power_Sphere | 0 | 22_Parallel_Extension | 48 | 33_Inferior_Pincer | 9 |

Table 8: Grasp type distribution across task categories: pick-and-place (T1), handover (T2), and affordance usage (T3).

| Grasp Type | T1 | T2 | T3 |
|---|---|---|---|
| 01_Large_Diameter | 23 | 15 | 11 |
| 02_Small_Diameter | 5 | 5 | 2 |
| 03_Medium_Wrap | 2 | 1 | 2 |
| 04_Adducted_Thumb | 15 | 5 | 4 |
| 05_Light_Tool | 0 | 1 | 3 |
| 06_Prismatic_4_Finger | 3 | 0 | 0 |
| 07_Prismatic_3_Finger | 2 | 0 | 2 |
| 08_Prismatic_2_Finger | 0 | 0 | 1 |
| 09_Palmar_Pinch | 1 | 1 | 1 |
| 10_Power_Disk | 4 | 1 | 0 |
| 12_Precision_Disk | 9 | 7 | 2 |
| 13_Precision_Sphere | 0 | 1 | 0 |
| 14_Tripod | 1 | 5 | 4 |
| 15_Fixed_Hook | 0 | 0 | 1 |
| 16_Lateral_Type | 2 | 4 | 0 |
| 17_Index_Finger_Extension | 9 | 4 | 6 |
| 18_Extension_Type | 5 | 10 | 2 |
| 22_Parallel_Extension | 19 | 17 | 12 |
| 23_Adduction_Grip | 0 | 0 | 2 |
| 24_Tip_Pinch | 0 | 0 | 1 |
| 27_Quadpod | 0 | 9 | 1 |
| 28_Sphere_3_Finger | 0 | 2 | 1 |
| 29_Stick | 3 | 1 | 4 |
| 30_Palmar | 17 | 13 | 3 |
| 31_Ring | 1 | 4 | 0 |
| 32_Ventral | 2 | 1 | 5 |
| 33_Inferior_Pincer | 4 | 4 | 1 |
| # Grasp Types | 19 | 21 | 22 |

examine their distribution across pick-and-place (T1), handover (T2), and affordance-usage (T3) tasks. The results reveal the richness and realism of everyday hand–object manipulation patterns captured in HO-Cap: T3 exhibits the greatest variety of grasp types, followed by T2 and T1, reflecting the increasing complexity of interactions across task categories.

# D    Comparison to Similar Datasets

The most related datasets to HO-Cap are DexYCB[8], HO-3D[21], and H2O [28]. **DexYCB** focuses on 20 objects from the YCB benchmark, primarily capturing single-hand grasping and handover interactions using 8 camera views. In contrast, HO-Cap provides 64 unique objects with textured 3D meshes, offering greater object diversity. Additionally, HO-Cap includes both single-hand and

Table 9: Statistics of the evaluation setup for seen object detection.

| Split | #Sub | #Obj | #View | #Seq | #Image | #Obj Anno |
|-------|------|------|-------|------|--------|-----------|
| Train | 9 | 64 | 8 | 48 | 36,295 | 177,564 |
| Val | 9 | 64 | 8 | 48 | 9,073 | 44,535 |
| Test | 8 | 64 | 8 | 16 | 12,336 | 62,598 |

bimanual interactions and incorporates egocentric (FPV) data, enhancing its applicability for AR/VR and human-centered AI research. **HO-3D** features grasping-centric hand poses with 10 objects, captured from 1–5 views per sequence, and relies on marker-based motion capture for annotations. In contrast, HO-Cap supports a significantly wider range of hand poses and grasps across 64 objects and employs a semi-automatic optimization-based annotation pipeline that does not require markers. **H2O** is one of the few datasets that support bimanual hand-object interactions while providing 6D object poses and 3D hand poses. However, it is limited to 8 objects. In comparison, HO-Cap extends this to 64 objects, enabling a wider range of bimanual grasping scenarios and more diverse hand-object interactions.

# E    Experimental Details

We benchmarked three tasks using our dataset: hand pose estimation, object detection, and novel object pose estimation. To enhance evaluation efficiency, all methods were tested on sampled keyframes from the dataset. All experiments were implemented using PyTorch and conducted on a desktop workstation equipped with a single Intel(R) Core(TM) i9-10900X CPU, dual NVIDIA RTX A5000 GPUs, 64 GB of RAM, and running Ubuntu 20.04.

For hand pose estimation, the evaluation frames were subsampled at 10 frames per second (FPS) across all eight RealSense cameras, resulting in 189,435 frames. This subset is sufficiently representative to capture the diversity of hand poses in the dataset. When computing the mean per joint position error (MPJPE), predictions and ground truths are aligned by replacing the root (wrist) location with the ground truth, eliminating translational ambiguity.

For 6D object pose estimation, due to significant occlusions at the two lowest camera angles and the egocentric perspectives dominated by hand views, we selected 11,758 frames from the remaining six viewpoints for evaluation. These chosen frames encompass diverse poses of objects. Ground truth bounding boxes were provided as input for the pose estimation methods.

For 2D object detection, evaluation frames were sampled from the RealSense camera feeds. The YOLO11 model was trained from scratch using the YOLO11m configuration, while RT-DETR was trained under default setting. For novel object detection, a random sampling strategy was employed, yielding 7,293 frames. Table 9 provides the statistics of the evaluation setup for seen object detection. Full frames were sampled across all eight RealSense cameras with a subsampling factor of 10 and divided into train/val/test splits. For each split, we list the number of subjects ("#sub"), objects ("#obj"), views ("#view"), sequences ("#seq"), image samples ("#image"), and the object annotations ("#obj anno").

# F    Qualitative: 3D Hand Pose Estimation

Fig. 9 shows qualitative results of 3D hand pose estimation using HaMeR [42] and A2J-Transformer [25] methods. As shown in the figure, the results include both 2D hand joints and 3D hand joints. HaMeR demonstrates superior accuracy in hand pose estimation compared to A2J-Transformer. Due to the limitations of the A2J-Transformer training process, it lacks consideration for the interaction between the hand and surrounding objects. The results show that as the interaction area between the hand and the object increases or when the hand is occluded by the object, the performance of A2J-Transformer deteriorates. Conversely, HaMeR exhibits a robust adaptability to these challenging conditions by training a ViT model with large-scale training images.

## G   Qualitative: Novel Object Detection

Fig. 10 shows qualitative results of 2D novel object detections with GroundingDINO [32] and CNOS [40]. For GroundingDINO, we used combined object product names as prompt captions. For CNOS, feature templates were created for each object by rendering textured 3D models from multiple viewpoints. Both methods exhibit a significant number of false positives when tested on our dataset. Given that each scene contains only four objects, Fig. 10 highlights the top four detected bounding boxes with the highest detection scores.

## H   Qualitative: 6D Object Pose Estimation

Fig. 11 shows qualitative results of 6D object pose estimation. The object models, rendered using the estimated poses, are overlaid onto a darkened input image for visualization. As shown, FoundationPose[52] generates more accurate 6D pose predictions compared to MegaPose [29] on novel objects.

## I   Quantitative: 6D Object Pose Estimation

For 6D object pose estimation, we include more detailed results for the evaluation metrics—namely, Average Distance (ADD) and Symmetric Average Distance (ADD-S)—on a per-object basis in Table 10. The relationships of objects and their IDs can be found in Fig. 12. We observe that FoundationPose [52] significantly surpasses MegaPose [29] in handling novel objects, demonstrating a substantial improvement in accuracy.

## J   Societal Impact

The HO-Cap dataset contributes to advancements in embodied AI, robotics, and augmented reality. By enabling research on hand-object manipulation, it supports applications in assistive robotics for people with disabilities, more natural human-computer interaction, and better simulation of human dexterity in learning systems. To protect privacy and minimize potential risks, the dataset does not contain facial data, speech, or biometric identifiers. We strongly encourage the responsible use of this dataset in accordance with the terms outlined in our dataset license, and discourage any applications that could lead to surveillance, profiling, or other unethical uses.

## K   License

The HO-Cap dataset is released under the Creative Commons Attribution-NonCommercial 4.0 International (CC BY-NC 4.0) license. This license permits non-commercial use, sharing, and adaptation with appropriate attribution.

The BundleSDF [51] and FoundationPose [52] components are distributed under the NVIDIA Source Code License, which restricts usage to research and academic purposes only.

The SAM2 [43] image and video prediction code, along with the MediaPipe [35] hand landmark prediction code, are licensed under the Apache License 2.0, which allows for broad use, modification, and distribution under open-source terms.

Table 10: 6D object pose estimation results of representative approaches in ADD and ADD-S.

| Object ID | FoundationPose [52] | | MegaPose [29] | | Object ID | FoundationPose [52] | | MegaPose [29] | |
|---|---|---|---|---|---|---|---|---|---|
| | ADD | ADD-S | ADD | ADD-S | | ADD | ADD-S | ADD | ADD-S |
| G01_1 | 93.66 | 96.00 | 88.90 | 93.59 | G11_1 | 89.40 | 95.74 | 66.49 | 82.61 |
| G01_2 | 93.55 | 96.18 | 80.08 | 88.33 | G11_2 | 89.43 | 95.74 | 66.20 | 82.51 |
| G01_3 | 92.98 | 95.96 | 71.38 | 81.96 | G11_3 | 89.39 | 95.67 | 65.92 | 82.62 |
| G01_4 | 88.39 | 95.58 | 69.44 | 82.26 | G11_4 | 89.42 | 95.67 | 65.45 | 82.21 |
| G02_1 | 89.60 | 95.72 | 69.37 | 82.89 | G15_1 | 89.43 | 95.67 | 65.37 | 82.18 |
| G02_2 | 90.19 | 95.79 | 71.22 | 84.50 | G15_2 | 89.41 | 95.67 | 65.33 | 82.18 |
| G02_3 | 90.82 | 95.93 | 71.72 | 85.56 | G15_3 | 89.42 | 95.67 | 65.31 | 82.18 |
| G02_4 | 91.00 | 95.83 | 70.88 | 85.11 | G15_4 | 89.53 | 95.68 | 65.28 | 82.10 |
| G04_1 | 91.10 | 95.85 | 69.84 | 84.39 | G16_1 | 89.57 | 95.69 | 65.56 | 82.14 |
| G04_2 | 91.25 | 95.88 | 69.17 | 84.33 | G16_2 | 89.65 | 95.70 | 65.57 | 82.18 |
| G04_3 | 91.29 | 95.87 | 69.33 | 84.37 | G16_3 | 89.67 | 95.69 | 65.42 | 81.97 |
| G04_4 | 91.36 | 95.87 | 70.08 | 84.73 | G16_4 | 89.72 | 95.70 | 65.57 | 82.03 |
| G05_1 | 90.54 | 95.90 | 69.25 | 84.79 | G18_1 | 89.67 | 95.61 | 65.41 | 82.05 |
| G05_2 | 90.86 | 95.92 | 69.14 | 84.76 | G18_2 | 89.70 | 95.61 | 65.48 | 82.04 |
| G05_3 | 90.82 | 95.94 | 69.21 | 84.80 | G18_3 | 89.72 | 95.61 | 65.68 | 82.13 |
| G05_4 | 86.23 | 95.90 | 66.57 | 85.10 | G18_4 | 89.77 | 95.59 | 65.92 | 82.25 |
| G06_1 | 87.28 | 95.94 | 67.48 | 85.33 | G19_1 | 89.75 | 95.57 | 66.25 | 82.32 |
| G06_2 | 88.13 | 95.99 | 66.02 | 84.59 | G19_2 | 88.11 | 95.59 | 65.90 | 81.72 |
| G06_3 | 88.42 | 96.01 | 66.97 | 84.84 | G19_3 | 87.67 | 95.56 | 64.78 | 81.61 |
| G06_4 | 88.41 | 95.83 | 65.75 | 82.95 | G19_4 | 87.70 | 95.57 | 65.09 | 81.75 |
| G07_1 | 88.52 | 95.81 | 65.58 | 82.50 | G20_1 | 87.76 | 95.57 | 64.57 | 81.79 |
| G07_2 | 88.57 | 95.81 | 65.65 | 82.52 | G20_2 | 87.83 | 95.58 | 64.64 | 81.88 |
| G07_3 | 88.72 | 95.84 | 66.04 | 82.66 | G20_3 | 87.59 | 95.58 | 64.89 | 82.01 |
| G07_4 | 88.78 | 95.81 | 66.14 | 82.50 | G20_4 | 87.64 | 95.59 | 64.30 | 81.84 |
| G09_1 | 88.96 | 95.71 | 67.04 | 82.84 | G21_1 | 87.73 | 95.60 | 64.40 | 81.99 |
| G09_2 | 89.20 | 95.68 | 68.11 | 83.26 | G21_2 | 87.79 | 95.59 | 64.68 | 82.15 |
| G09_3 | 89.51 | 95.73 | 67.68 | 83.27 | G21_3 | 87.68 | 95.56 | 64.29 | 81.63 |
| G09_4 | 89.72 | 95.75 | 67.51 | 83.34 | G21_4 | 87.65 | 95.56 | 63.87 | 81.13 |
| G10_1 | 89.82 | 95.77 | 67.64 | 83.44 | G22_1 | 87.63 | 95.57 | 64.01 | 81.23 |
| G10_2 | 89.81 | 95.78 | 67.58 | 83.48 | G22_2 | 87.69 | 95.57 | 63.68 | 81.13 |
| G10_3 | 89.88 | 95.78 | 67.72 | 83.51 | G22_3 | 87.73 | 95.58 | 63.64 | 81.12 |
| G10_4 | 89.52 | 95.76 | 66.92 | 83.03 | G22_4 | 89.33 | 95.74 | 63.53 | 80.97 |

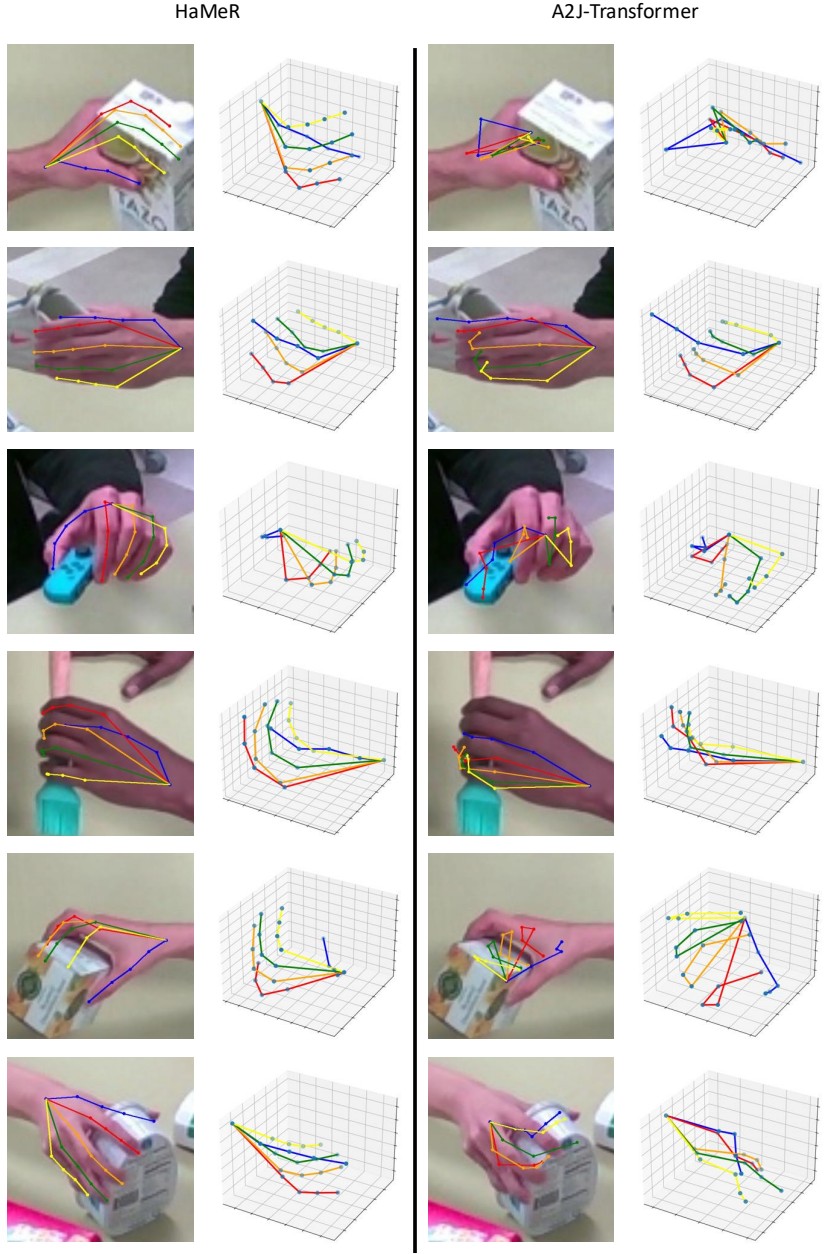

Figure 9: Qualitative results of the predicted 3D hand pose using HaMeR [42] (left two columns) and A2J-Transformer [25] (right two columns).

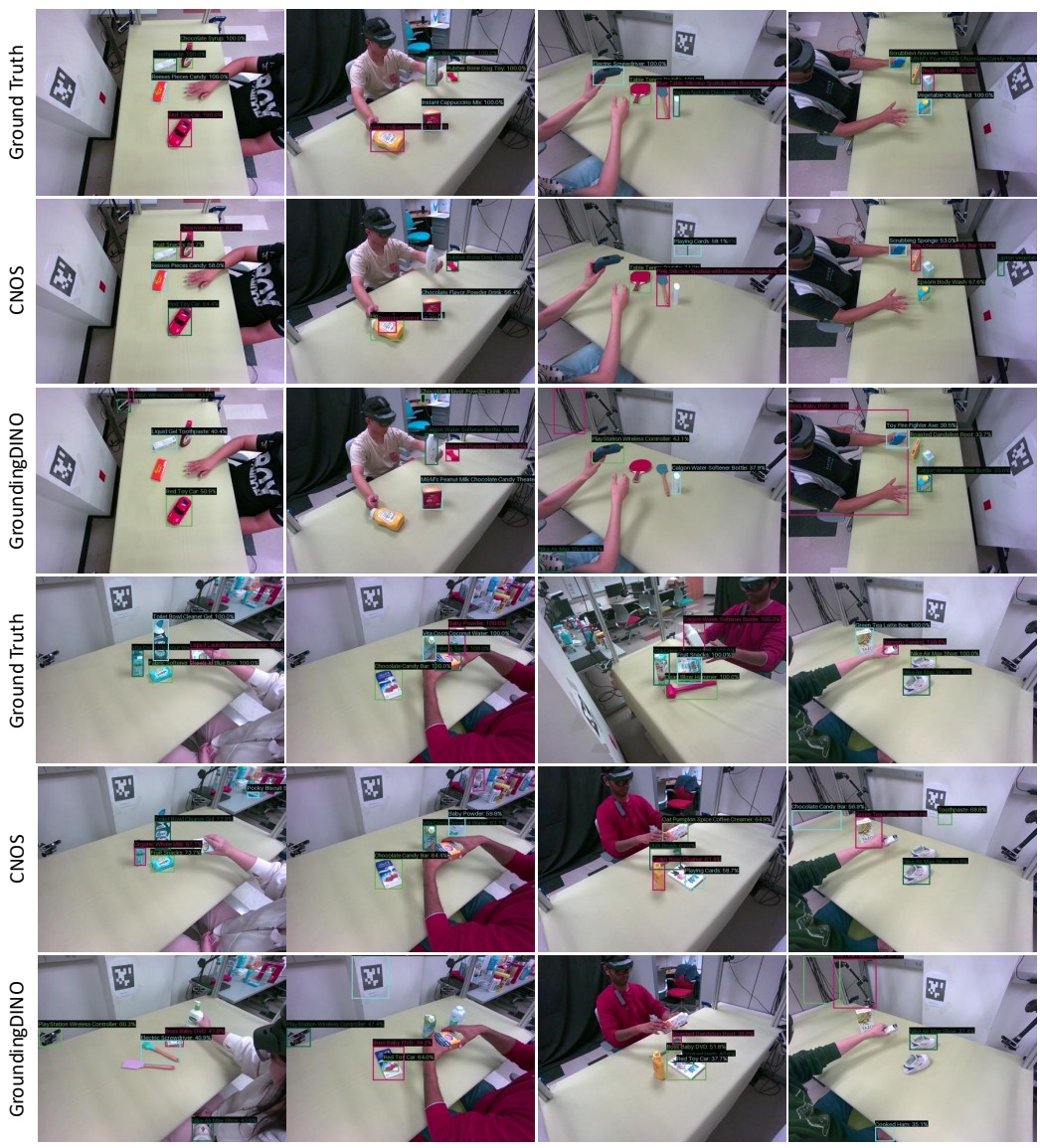

Figure 10: Qualitative results for novel object detection

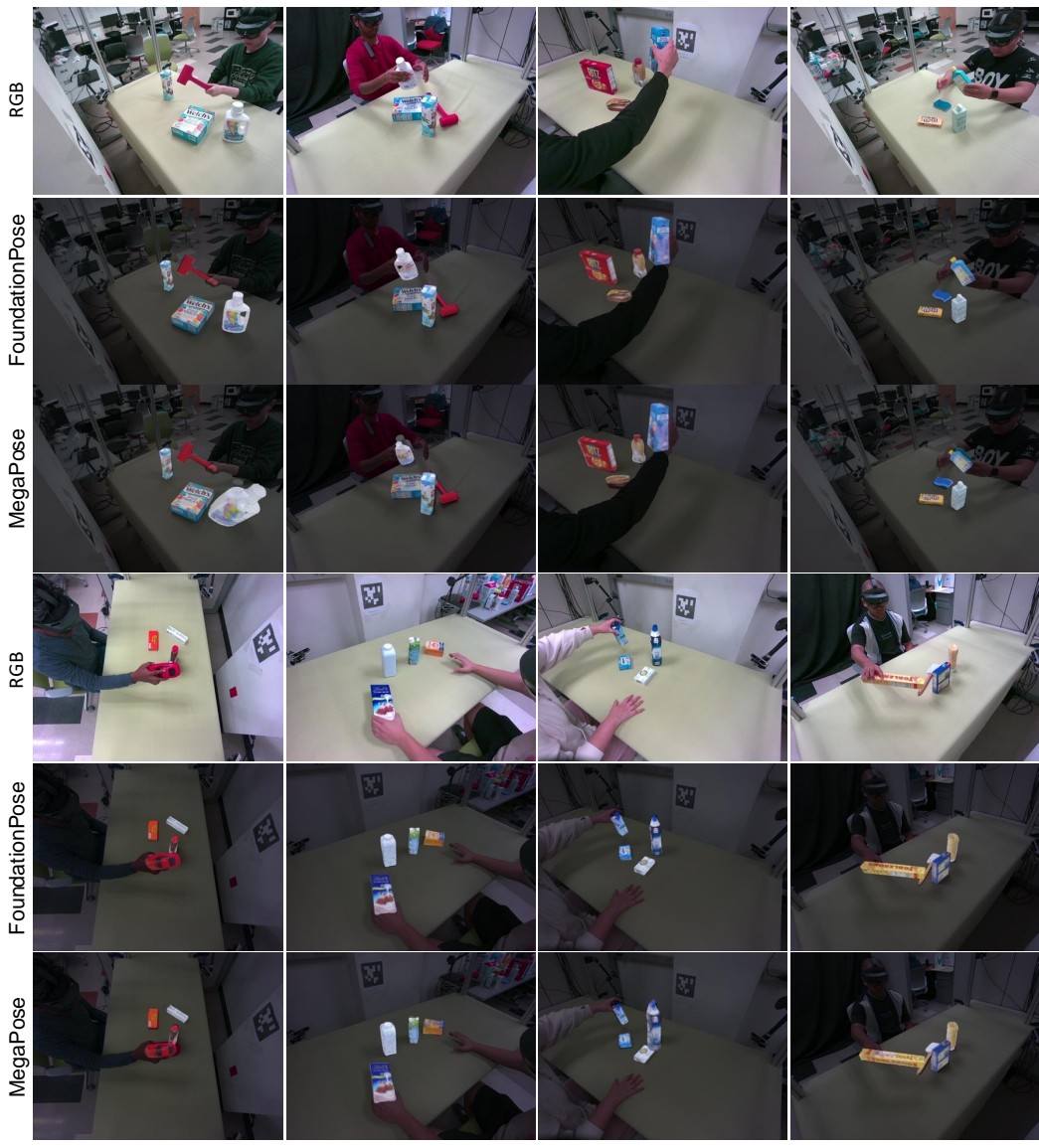

Figure 11: Qualitative results of 6D object pose estimation. (Top to down: Input RGB frame, FoundationPose [52], MegaPose [29])

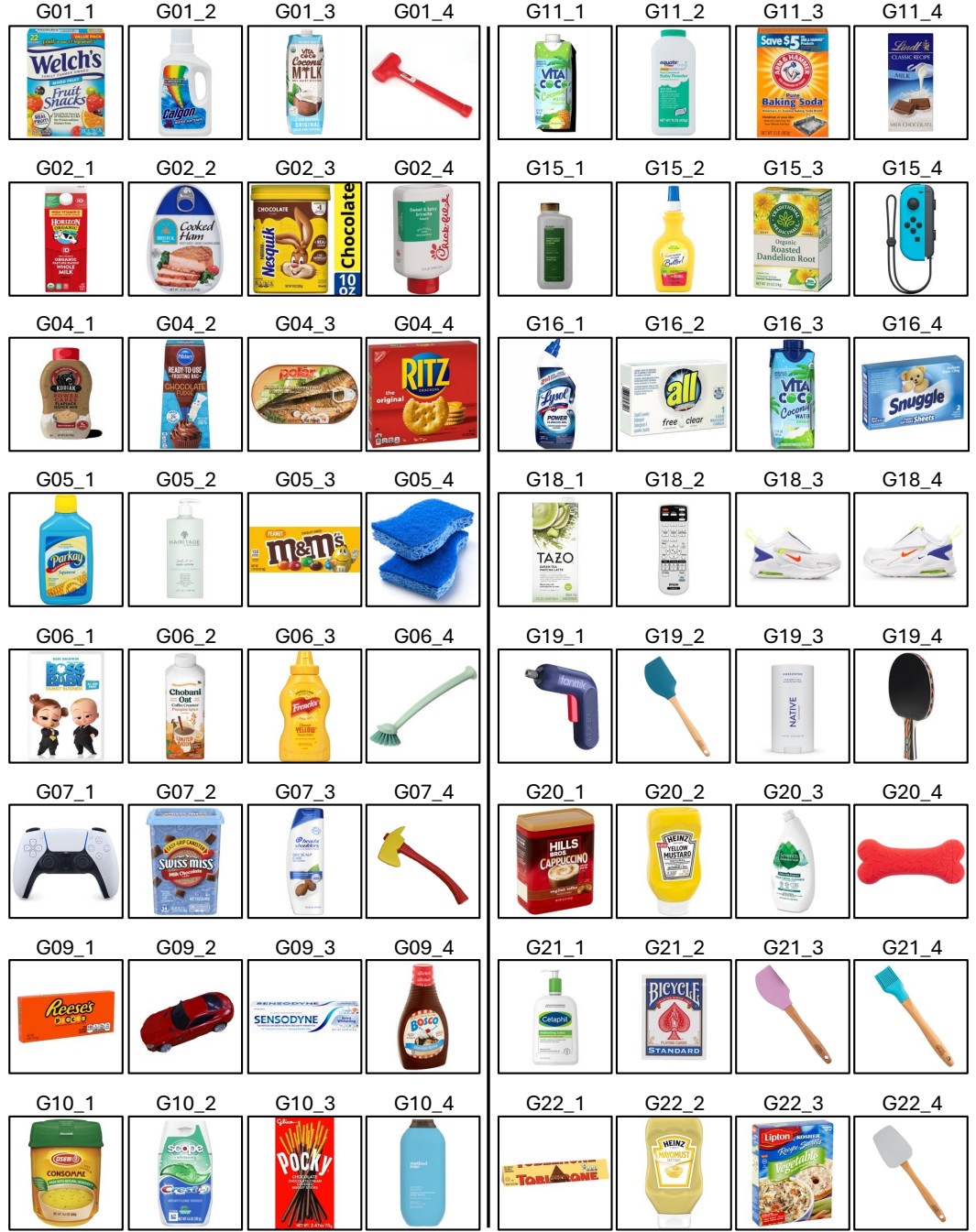

Figure 12: Objects with their IDs in our dataset.

