# OpenReview forum: "HO-Cap: A Capture System and Dataset for 3D Reconstruction and Pose Tracking of Hand-Object Interaction"
_NeurIPS.cc/2025/Datasets_and_Benchmarks_Track — NeurIPS 2025 Datasets and Benchmarks Track poster_

### Official Review · Reviewer_16sU · 2025-06-24

**Rating:** 4
**Confidence:** 4

**Summary:**

The paper introduces HO-Cap, a new dataset and data capture pipeline for 3D hand-object interaction, leveraging powerful off-the-shelf models such as SAM2 and FoundationPose. The system combines an eight-camera multi-view setup with an egocentric HoloLens headset, integrating both 3D hand-object optimization and model predictions. The dataset comprises 64 videos and 656K RGB-D frames, capturing 9 subjects interacting with 64 objects. Additionally, the paper evaluates off-the-shelf models on tasks including CAD-based object detection, open-vocabulary detection, hand pose estimation, and unseen object pose estimation.

**Additional Feedback:**

N/A

**Dataset Code Accessibility:**

Yes

**Dataset Code Comments:**

The authors provided link for the dataset, which is publicly available.

**Ethical Considerations:**

No, there are no or only very minor ethics concerns

**Final Justification:**

This paper introduces HO-Cap, a newly captured dataset for 3D hand–object reconstruction, with its claimed novelty primarily stemming from the data collection process. The method employs multiple off-the-shelf tools (e.g., MediaPipe) and foundational models (e.g., SAM2, FoundationPose), followed by optimization for hand and object. While this approach is novel, it inherits the limitations of the foundational models, whose accuracy and fidelity remain imperfect, necessitating further heuristic post-processing.

As shown in Table 3 of the main manuscript, when comparing manually annotated samples to those annotated by HO-Cap, the latter shows higher error which exceeds 14 mm for the left hand. Given this, the dataset might be more accurately described as pseudo-GT rather than GT if the paper’s description were more concrete. The significance of a dataset in this field depends on unique characteristics that distinguish it from existing ones; however, many reviewers seemed to struggle to identify such distinguishing traits here.

While my rating has been updated as the authors addressed one of my key concerns, I believe the paper still has room for improvement. Therefore, I remain within Borderline ratings.

**Limitations Weaknesses:**

1. Authors claim that annotation costs are high due to 3D scanners or motion capture systems, yet their method also relies on a multi-view camera setup with triangulation and optimization, similar to existing datasets like DexYCB and others in Table 1. What concrete advantages does HO-Cap offer over these methods? Can the authors provide direct quantitative comparisons in terms of accuracy and labeling efficiency on HO-Cap evaluation set between HO-Cap's annotation process and existing method's annotation process?
2. What added value does HO-Cap provide as a dataset? New datasets typically expand coverage of attributes like novel objects, hand poses, novel actions, or in-the-wild scenarios. HO-Cap appears limited in terms of this direction. What aspect of HO-Cap dataset would it contribute as an additional resource?
3. The use of models like FoundationPose, SAM2, and MediaPipe followed by joint optimization seems better suited for in-the-wild scenarios with difficult calibration. Given that HO-Cap is captured in controlled environments, is this level of complexity necessary?
4. How does HO-Cap's flexibility regarding camera calibration compare to existing motion capture systems like DexYCB or OakInk? Does it meaningfully reduce calibration constraints?
5. The main hardware difference from DexYCB appears to be HoloLens. How significantly does it improve accuracy or ease of data capture?
6. Comparing HaMeR to InterWild (Moon et al., CVPR 2023) would be more appropriate than A2J-Transformer, as InterWild is also trained on in-the-wild images, unlike A2J-Transformer which has limited performance on unseen images.

**Strengths Contributions:**

1. The paper offers a useful overview and comparison of existing models for 3D hand pose estimation (Table 4), object detection (Table 5), and 6D object pose estimation (Table 6) in the context of 3D hand-object data capture.
2. The dataset acquisition process requires minimal prerequisites, notably eliminating the need for CAD models of each object, which facilitates scalable data collection.

---

> ### Author Rebuttal · Authors · 2025-07-30
>
> > **Q1 (Summary): How is HO-Cap more efficient than existing multi-view methods like DexYCB, and can its annotation accuracy and cost be quantitatively compared?**
>
> **A1:**
> Our HO-Cap annotation pipeline provides following key advantages:
>
> - Cost-efficient and Hardware-light: The pipeline eliminates the need for manual labeling and expensive equipment such as motion capture systems or 3D scanners. Instead, it relies solely on affordable, consumer-grade RGB-D cameras (e.g., Intel RealSense), without requiring external tracking hardware or labor-intensive annotation processes.
> - Easily Extensible to New Object Categories and Hand Shapes: our pipeline does not depend on pre-scanned object meshes and fixed object libraries. The object shape collection and MANO shape calibration procedures are straightforward, enabling easy extension to new object categories and hand shapes with minimal overhead.
> - High Annotation Accuracy: As demonstrated in Table 3, our SDF-based optimization achieves annotation accuracy comparable to manual labeling, with light error margins for both hand and object poses.
>   In summary, HO-Cap offers a cost-effective, easily scalable, and high-quality annotation pipeline that significantly lowers the barrier to hand-object dataset collection. We believe this will broaden participation in hand-object research and encourage wider adoption and community contributions.
>
> Regarding the reviewer’s request for direct quantitative comparisons in terms of annotation accuracy and labeling efficiency, we fully agree this is important. We are currently running our pipeline on DexYCB, which has a similar hardware setup, and will report detailed comparisons in the final supplementary material.
>
> > **Q2: What added value does HO-Cap provide as a dataset? New datasets typically expand coverage of attributes like novel objects, hand poses, novel actions, or in-the-wild scenarios. HO-Cap appears limited in terms of this direction. What aspect of HO-Cap dataset would it contribute as an additional resource?**
>
> **A2:**
> We appreciate the reviewer’s question and would like to highlight several key aspects that distinguish HO-Cap as a valuable and complementary resource for the community:
>
> - **Joint Egocentric + Third-Person Multi-View Setup:** HO-Cap provides synchronized egocentric (HoloLens) and third-person multi-view RGB-D recordings. This enables research on cross-view learning, egocentric perception, and domain adaptation across perspectives, which are not supported by prior datasets like DexYCB, HO3D, or OakInk.
> - **Rich and Dense Interaction Scenarios:** In contrast to datasets focused on single-object grasping or short interactions, HO-Cap captures multi-object rearrangement, tool usage simulation, and bimanual hand-object interactions. These sequences feature long-duration contact, frequent occlusions, and task-driven manipulation, offering a more realistic and challenging benchmark for model development.
> - **Scalable, Model-Agnostic Annotation Pipeline:** HO-Cap supports rapid expansion to novel objects and unseen hand shapes without requiring pre-existing CAD models or scanning. This makes it ideal for few-shot and open-set learning tasks, and encourages future community-driven dataset growth.
> - **Integration of Foundation and Neural Models:** HO-Cap is among the first datasets to integrate SAM2, FoundationPose, MediaPipe, and BundleSDF into a unified annotation pipeline, operating under model-free, occlusion-rich, real-world settings. It serves as a testbed for evaluating and extending foundation models within hand-object understanding pipelines, a growing trend in the community.
>
> In summary, while HO-Cap is not positioned as an in-the-wild dataset, it uniquely offers a realistic, scalable, and well-suited platform for foundation model integration, enabling advances in hand-object interaction research across both egocentric and third-person views.
>
> > **Q3: The use of models like FoundationPose, SAM2, and MediaPipe followed by joint optimization seems better suited for in-the-wild scenarios with difficult calibration. Given that HO-Cap is captured in controlled environments, is this level of complexity necessary?**
>
> **A3:**
> We appreciate the reviewer’s question. The controlled environments are necessary for HO-Cap:
>
> - **Multi-view input is essential for SDF-based optimization:** Our pipeline relies on a dense 3D point cloud of the scene, which is constructed from multiple synchronized RGB-D views. This full-scene reconstruction is critical for pose optimization with signed distance fields and cannot be reliably achieved with single-view input.
> - **Single-view predictions degrade under occlusion:** In scenarios with significant hand-object occlusion, FoundationPose and SAM2 often lose tracking in a single view. MediaPipe, in particular, is known to predict incorrect landmarks when hands are partially or fully occluded.
> - **Multi-view observations improve robustness:** By leveraging eight camera views, we significantly increase the chances that the target object and hands remain visible in at least some views during occlusion-heavy interactions. This visibility enables more stable detection, and when combined with RANSAC-based filtering and multi-view fusion, leads to much higher annotation success rates.
>
> In summary, although our environment is controlled, the complexity of the pipeline is not excessive but essential to achieve high-quality, robust annotation in natural, occlusion-rich, task-driven hand-object interaction sequences.
>
> > **Q4: How does HO-Cap's flexibility regarding camera calibration compare to existing motion capture systems like DexYCB or OakInk? Does it meaningfully reduce calibration constraints?**
>
> **A4:**
> We appreciate the reviewer’s question. HO-Cap offers significant flexibility in terms of camera calibration and setup, especially when compared to motion capture systems and existing multi-camera datasets like DexYCB. This flexibility comes from the following aspects:
>
> **Compared to motion capture systems (e.g., FPHA, ContactPose, ARCTIC, OakInk2):**
>
> - **Cost-Efficient Setup:** HO-Cap relies on affordable, off-the-shelf RGB-D cameras (e.g., Intel RealSense), which are easy to set up and configure. In contrast, mocap systems require multiple high-precision motion capture cameras, each of which can be several times more expensive than a RealSense camera.
> - **Simplified Calibration:** HO-Cap uses a standard checkerboard-based calibration method to estimate intrinsic and extrinsic parameters across multiple views. This process is straightforward, repeatable, and does not require specialized rigs or software. By contrast, mocap systems require precise alignment between marker coordinates and image cameras, which involves complex calibration procedures and custom hardware. They are also prone to alignment artifacts due to occlusions or coordinate mismatches.
> - **Reduced Annotation Overhead:** Mocap systems require careful and consistent marker placement on hands and objects, which is both time-consuming and error-prone, especially in contact-rich manipulation scenes. HO-Cap avoids this entirely by relying on vision-based models and multi-view geometry.
>
> **Compared to DexYCB (a multi-camera, markerless dataset):**
>
> - **Greater Flexibility for Object Expansion:** HO-Cap is easy to extend to new object categories without extra hardwares using a lightweight reconstruction pipeline. In contrast, DexYCB relies on predefined YCB object meshes and requires manual labeling or object-specific scanning, which limits scalability and increases labor cost.
> - **Low Labor Overhead:** DexYCB requires extensive manual annotation for each object and hand, which is labor-intensive and time-consuming. HO-Cap’s pipeline automates most of the annotation process, significantly reducing the time and effort required to collect new sequences.
>
> In summary, HO-Cap significantly reduces calibration and setup constraints while maintaining annotation quality, making it more extensible, and accessible for research and future dataset collection.
>
> > **Q5: The main hardware difference from DexYCB appears to be HoloLens. How significantly does it improve accuracy or ease of data capture?**
>
> **A5:**
> While the primary hand and object pose annotations in HO-Cap are indeed optimized using the 8 static RealSense cameras, the HoloLens serves a complementary and valuable role by providing egocentric RGB data. This perspective is particularly important for:
>
> - AR/VR and robotics applications, where first-person hand-object interactions and occlusions are common.
> - Supporting egocentric perception research, which is underserved in existing datasets like DexYCB.
> - Provide annotations for cross-view consistency learning by aligning egocentric and third-person observations.
>
> Although it does not directly improve annotation accuracy, it enriches the dataset’s applicability and realism, helping bridge the gap between controlled lab data and real-world, human-centric scenarios.
>
> > **Q6: Comparing HaMeR to InterWild (Moon et al., CVPR 2023) would be more appropriate than A2J-Transformer, as InterWild is also trained on in-the-wild images, unlike A2J-Transformer which has limited performance on unseen images.**
>
> **A6:**
> We thank the reviewer for the suggestion. We agree that InterWild is a more appropriate baseline for HaMeR. We will include a comparison with InterWild on Hand Pose Evaluation and update Table 4 accordingly.
>
> | Method    | PCK@0.5 $\uparrow$ | PCK@0.1 $\uparrow$ | PCK@0.15 $\uparrow$ | PCK@0.2 $\uparrow$ | MPJPE (mm) $\downarrow$ |
> | --------- | ------------------ | ------------------ | ------------------- | ------------------ | ----------------------- |
> | HaMeR     | 43.7               | 79.2               | 88.5                | 91.4               | 28.9                    |
> | InterWild | 51.7               | 60.9               | 70.0                | 78.6               | 57.6                    |

---

> > ### Comment · Reviewer_16sU · 2025-08-03
> >
> > First, thank you for the responses by authors. Below are the key points that still seem insufficiently addressed in the paper and the rebuttal:
> >
> >
> > ## Limited Advantage over DexYCB
> >
> > **SAM2 and FoundationPose are PoseCNN Replacements**
> >
> > The use of SAM2 and FoundationPose is analogous to PoseCNN in DexYCB, which provides initial pose estimates. Both pipelines still rely on optimization for refinement in the end. The usage of foundation models for initialization seems more of an initialization with current SOTA methods than a fundamental shift. While HO-Cap avoids pre-scanned object templates, it results in low-quality meshes (Fig. 7), limiting its value for 3D reconstruction tasks like MOHO [1].
> >
> >
> > **Ambiguity in Table 3**
> >
> > It’s unclear whether the annotated data from HO-Cap with the proposed method has comparable accuracy to manual annotations. Still, the error is high compared to the manual annotation (assuming that Table 3 compared the proposed method with manual annotation). If there is high labeling error compared to manual annotation, then wouldn't it negatively impact the fidelity of the collected data as proper dataset?
> >
> >
> > ## Pseudo-GT Rather Than True GT
> > Given the low fidelity of reconstructed meshes (Fig. 7), HO-Cap serves more as pseudo ground truth than actual GT. Since object scanning isn’t the main bottleneck from previous methods, why not include it to ensure quality?
> >
> > ## Dataset Contributions Need Clarification
> >
> > **Egocentric + Third-Person Views**
> >
> > This setup has been explored in prior datasets (e.g., H2O, ARCTIC, OakInk2), so there seems to be no added knowledge as a dataset in this direction. Is there something that I am missing?
> >
> >
> > **Rich Interaction Scenarios**
> >
> > This part is interesting but claims of rich interactions are not properly illustrated in the paper. Only Table 2 touches on this briefly. Please provide a more concrete statistics (e.g., two-hand interactions) would help differentiate HO-Cap rather than vague words like "task-driven" or "high-quality" to make this statement more convincing.
> >
> >
> > **Scalable, Model-Agnostic Pipeline**
> >
> > While scalable, the pipeline yields poor mesh quality (especially the object meshes), reducing utility for key 3D tasks. Why should people still use this dataset for training their model?
> >
> >
> > **Foundation Models + Optimization**
> > Since the foundation models (SAM2, MediaPipe, FoundationPose) are error-prone (L184-L185, L321, L141-L143), optimization remains essential. What practical benefit do these models offer if they must be corrected post-hoc?
> >
> >
> > ## Tradeoff with Motion Capture
> > Ease of annotation seems to come at the cost of accuracy. Compared to traditional mocap, the technical reliability of this pipeline is still lacking.
> >
> >
> > The InterWild results are appreciated and clearly stronger than A2J-Transformer.
> >
> >
> > [1] Zhang et al., MOHO: Learning Single-view Hand-held Object Reconstruction with Multi-view Occlusion-Aware Supervision, CVPR 2024

---

> > > ### Author Response · Authors · 2025-08-03
> > >
> > > We thank the reviewer for the thoughtful feedback. Below, we address each point in detail:
> > >
> > > ## Limited Advantage over DexYCB
> > >
> > > HO-Cap offers several advantages over DexYCB:
> > >
> > > - **Foundation Models over PoseCNN:** SAM2 and FoundationPose generalize across diverse objects categories, unlike PoseCNN, which requires per-object training on extral labeled data. HO-Cap automates per-frame initialization using these models, while DexYCB relies on manually selecting poses (as noted in their paper), limiting scalability.
> > > - **Open Source:** HO-Cap is fully open-sourced, enabling broad reuse and community extension.
> > > - **Mesh Quality Trade-off:** Our goal is to support efficient annotation and interaction modeling, not photorealistic surface reconstruction. While neural-reconstructed meshes are lower in fidelity, they are sufficient for generating physically plausible hand-object poses, segmentation masks, and interaction analysis. High-resolution scans are not the pipeline’s bottleneck; for tasks requiring high-res geometry (e.g., MOHO), we encourage the community to contribute higher-quality meshes. Our framework remains compatible with external scanned models.
> > > - **Annotation Accuracy:** As shown in Table 3, our pixel-level accuracy is comparable to DexYCB. Although the 3D error is slightly higher due to unrefined triangulated ground truth, it remains acceptable when compared to H2O.
> > >
> > > ## Pseudo-GT Rather Than True GT
> > >
> > > We intentionally avoid object scanning to maintain scalability. Scanning is time-consuming, hardware-dependent, and limits extensibility. Instead, HO-Cap leverages neural reconstruction and foundation models to reduce annotation bottlenecks across the entire pipeline—including shape, pose, and interaction — without requiring object-specific supervision. While the mesh fidelity is lower than scanned models, it is sufficient for pose refinement, contact evaluation, and interaction modeling, which are the core goals of HO-Cap. For downstream tasks requiring high-res geometry (e.g., MOHO), we welcome community contributions of higher-quality meshes. Our pipeline remains compatible with external scanned models.
> > >
> > > ## Dataset Contributions Need Clarification
> > >
> > > We are pleased to clarify the contributions of HO-Cap in the following key areas:
> > >
> > > - **Egocentric + Third-Person Views:** While similar setups exist, HO-Cap differs in key aspects:
> > >   - **HoloLens vs. Head-Mounted Cameras:** HO-Cap uses a HoloLens to capture egocentric RGB, providing a realistic first-person perspective without the artifacts associated with head-mounted cameras. This is crucial for AR/VR applications.
> > >   - **No Markers and Reflective Materials:** Unlike H2O, ARCTIC, and OakInk2, which use reflective materials or markers for calibration and tracking, HO-Cap captures egocentric RGB without such artifacts. This enables research on egocentric–exocentric domain adaptation, occlusion recovery, and view-consistent hand-object understanding, with cleaner input data.
> > > - **Rich Interaction Scenarios:** Although we define three general task types, participant behavior was unconstrained to promote diversity. This results in a wide range of interaction dynamics, including whole-hand and fingertip grasps, in-hand object manipulation, and bimanual interactions. We will release additional annotation and statistics in future dataset updates to better illustrate this meta-interaction diversity.
> > > - **Scalable, Model-Agnostic Pipeline:** Despite lower mesh fidelity, the HO-Cap pipeline:
> > >   - Avoids dependency on pre-scanned templates.
> > >   - Enables extension to new object categories and hand-object interaction tasks with minimal overhead.
> > >   - The HO-Cap dataset provides accurate hand-object pose annotations, segmentation masks, and interaction dynamics, which are sufficient for training and evaluating models on hand pose estimation, object pose extimation, hand-object interaction understanding, and contact analysis tasks, as well as for downstream applications like AR/VR, robotics, and human-robot interaction.
> > > - **Use of Foundation Models:** Alougth not perfect under specific conditions, foundation models significantly reduce manual effort, generalize well, and serve as effective priors for SDF-based optimization, ensuring robust annotations.
> > >
> > > ## Tradeoff with Motion Capture
> > >
> > > HO-Cap prioritizes scalability and accessibility over the sub-millimeter precision of traditional motion capture. By using only consumer-grade RGB-D sensors and foundation model priors—without markers or specialized hardware—our pipeline achieves competitive accuracy, as shown in Table 3 and interaction metrics (e.g., penetration depth, intersection volume). While not intended to replace mocap for ultra-high-precision tasks, HO-Cap provides physically plausible annotations in complex, occlusion-rich scenes, significantly lowering the cost and setup barriers for large-scale hand-object interaction capture.

---

> > > > ### Comment · Reviewer_16sU · 2025-08-05
> > > >
> > > > Thank you again for your response.
> > > >
> > > > The most important aspect of my previous comment was concerning the dataset contribution. In particular, I believe the “Rich Interaction Scenarios” is the most critical component, as it differentiates the HO-Cap dataset from existing datasets.
> > > >
> > > > Could you please provide a more detailed quantitative analysis and statistics to support this contribution? Without such analysis, it is difficult to assess the significance and impact of the dataset.

---

> > > > > ### Author Response · Authors · 2025-08-06
> > > > >
> > > > > We thank the reviewer again for the valuable feedback.
> > > > >
> > > > > To address the concern, we provide a quantitative analysis of the grasp types in the HO-Cap dataset, based on the 33 grasp taxonomies defined by Feix et al. [1].
> > > > >
> > > > > **By Grasp Type**
> > > > >
> > > > > HO-Cap includes 28 out of 33 grasp types, significantly more than HO-3D (15) and DexYCB (14), indicating broader grasp diversity. The count for each type reflects how many unique sequences include at least one instance.
> > > > >
> > > > > | Grasp Type            | #Sequences | Grasp Type                | #Sequences | Grasp Type         | #Sequences |
> > > > > | --------------------- | ---------- | ------------------------- | ---------- | ------------------ | ---------- |
> > > > > | 01_Large_Diameter     | 49         | 12_Precision_Disk         | 18         | 23_Adduction_Grip  | 2          |
> > > > > | 02_Small_Diameter     | 12         | 13_Precision_Sphere       | 1          | 24_Tip_Pinch       | 1          |
> > > > > | 03_Medium_Wrap        | 5          | 14_Tripod                 | 10         | 25_Lateral_Tripod  | 0          |
> > > > > | 04_Adducted_Thumb     | 24         | 15_Fixed_Hook             | 1          | 26_Sphere_4_Finger | 0          |
> > > > > | 05_Light_Tool         | 4          | 16_Lateral_Type           | 6          | 27_Quadpod         | 10         |
> > > > > | 06_Prismatic_4_Finger | 3          | 17_Index_Finger_Extension | 19         | 28_Sphere_3_Finger | 3          |
> > > > > | 07_Prismatic_3_Finger | 4          | 18_Extension_Type         | 17         | 29_Stick           | 8          |
> > > > > | 08_Prismatic_2_Finger | 1          | 19_Distal_Type            | 0          | 30_Palmar          | 33         |
> > > > > | 09_Palmar_Pinch       | 3          | 20_Writing_Tripod         | 0          | 31_Ring            | 5          |
> > > > > | 10_Power_Disk         | 5          | 21_Tripod_Variation       | 0          | 32_Ventral         | 8          |
> > > > > | 11_Power_Sphere       | 0          | 22_Parallel_Extension     | 48         | 33_Inferior_Pincer | 9          |
> > > > >
> > > > > **By Task Type**
> > > > >
> > > > > We further analyze how each grasp type appears across three task categories, each encompassing—but not limited to—the following actions:
> > > > >
> > > > > - **T1: Pick-and-Place:** Lifting and placing objects at the same or different locations, with or without object rotation.
> > > > > - **T2: Handover:** Transferring objects between hands, including bimanual and cooperative manipulation.
> > > > > - **T3: Affordance Usage:** Interacting with objects based on their functions (e.g., pouring, stirring, pressing, shaking, pushing, pulling, turning).
> > > > >
> > > > > Each cell indicates how many unique sequences in that task category contain the corresponding grasp type. The bottom row indicates how many distinct grasp types appear per task.
> > > > >
> > > > > | Grasp Type                | T1  | T2  | T3  |
> > > > > | ------------------------- | --- | --- | --- |
> > > > > | 01_Large_Diameter         | 23  | 15  | 11  |
> > > > > | 02_Small_Diameter         | 5   | 5   | 2   |
> > > > > | 03_Medium_Wrap            | 2   | 1   | 2   |
> > > > > | 04_Adducted_Thumb         | 15  | 5   | 4   |
> > > > > | 05_Light_Tool             | 0   | 1   | 3   |
> > > > > | 06_Prismatic_4_Finger     | 3   | 0   | 0   |
> > > > > | 07_Prismatic_3_Finger     | 2   | 0   | 2   |
> > > > > | 08_Prismatic_2_Finger     | 0   | 0   | 1   |
> > > > > | 09_Palmar_Pinch           | 1   | 1   | 1   |
> > > > > | 10_Power_Disk             | 4   | 1   | 0   |
> > > > > | 12_Precision_Disk         | 9   | 7   | 2   |
> > > > > | 13_Precision_Sphere       | 0   | 1   | 0   |
> > > > > | 14_Tripod                 | 1   | 5   | 4   |
> > > > > | 15_Fixed_Hook             | 0   | 0   | 1   |
> > > > > | 16_Lateral_Type           | 2   | 4   | 0   |
> > > > > | 17_Index_Finger_Extension | 9   | 4   | 6   |
> > > > > | 18_Extension_Type         | 5   | 10  | 2   |
> > > > > | 22_Parallel_Extension     | 19  | 17  | 12  |
> > > > > | 23_Adduction_Grip         | 0   | 0   | 2   |
> > > > > | 24_Tip_Pinch              | 0   | 0   | 1   |
> > > > > | 27_Quadpod                | 0   | 9   | 1   |
> > > > > | 28_Sphere_3_Finger        | 0   | 2   | 1   |
> > > > > | 29_Stick                  | 3   | 1   | 4   |
> > > > > | 30_Palmar                 | 17  | 13  | 3   |
> > > > > | 31_Ring                   | 1   | 4   | 0   |
> > > > > | 32_Ventral                | 2   | 1   | 5   |
> > > > > | 33_Inferior_Pincer        | 4   | 4   | 1   |
> > > > > | # Grasp Types             | 19  | 21  | 22  |
> > > > >
> > > > > **Summary**
> > > > >
> > > > > These results demonstrate HO-Cap’s rich variety of grasp types and realistic hand-object interaction scenarios, supporting its value over existing datasets.
> > > > >
> > > > > [1] Feix, T., et al. "The grasp taxonomy of human grasp types". IEEE Transactions on Human-Machine Systems (2015).

---

> > > > > > ### Comment · Reviewer_16sU · 2025-08-07
> > > > > >
> > > > > > I appreciate the authors' responses, which provided a clear quantitative analysis of the dataset’s contribution regarding the "Rich Interaction Scenarios." While the dataset may not be significantly superior to prior work in terms of overall contribution, the annotation process and its small yet unique properties are valuable to the field. Therefore, I will raise my rating to Borderline Accept, contingent on the inclusion of the presented statistics during the rebuttal in both the paper and the dataset annotation.

---

### Official Review · Reviewer_xudw · 2025-07-01

**Rating:** 5
**Confidence:** 5

**Summary:**

This paper proposes HO-Cap, a novel and cost-effective capture system for generating 3D hand-object interaction data without relying on traditional motion-capture systems. The authors introduce a semi-automatic annotation pipeline that combines multi-view video with reconstructed 3D meshes, leveraging Surface Distance Similarity (SDS) loss for aligning object pose with the fused point cloud. Additional refinements are applied to hand pose via 2D hand keypoint estimation and a 3D-to-2D projection-based correction, enhancing annotation quality. The approach is efficient and significantly reduces the cost and complexity of collecting hand-object interaction data.

**Additional Feedback:**

Since the data is estimated rather than captured via traditional mocap, it is expected to be more diverse and generalizable. However, the current set of activity types is quite limited. Does this reflect any limitations of the estimation method? Additionally, although the paper title emphasizes hand-object interaction, the interaction aspect is somewhat underrepresented in the main content.

**Dataset Code Accessibility:**

Yes

**Dataset Code Comments:**

Their dataset website is well-designed, and the dataset has been uploaded to HuggingFace, allowing for direct download.

**Ethical Comments:**

There are no significant ethical concerns identified in this work. The data collection process appears to follow standard protocols, and no sensitive or personally identifiable information is involved. The research focuses on technical advancements without introducing potential risks to individuals or society.

**Ethical Considerations:**

No, there are no or only very minor ethics concerns

**Final Justification:**

The authors clarified the reasons for the current dataset’s limited size. I encourage the authors to scale up the dataset in future iterations. These responses address my concern. I decided to raise my overall rating.

**Limitations Weaknesses:**

1. In the current loss design, hand and object are still optimised largely separately; the relation between them appears weak. Stronger hand–object interaction constraints (e.g., contact optimisation) could be explored.
2. The dataset contains only 64 videos, roughly one-tenth of OakInk2(Mocap). Since the annotation is advertised as lightweight, why not collect more data?
3. In the experiments, the authors treat hand-pose estimation and object-pose estimation as two separate tasks. However, given that the paper’s title emphasizes hand-object interaction, it lacks interaction-specific metrics—such as evaluating how well the hand and object surface meshes make contact within the dataset. Aside from briefly mentioning joint hand-object pose estimation, the method largely appears to tackle hand and object estimation independently.
4. Please add a comparison with the HOT3D[1] dataset.

[1] Banerjee, Prithviraj, et al. "Hot3d: Hand and object tracking in 3d from egocentric multi-view videos." Proceedings of the Computer Vision and Pattern Recognition Conference. 2025.

**Strengths Contributions:**

1.	Compared to traditional motion capture annotation methods, the approach proposed in this paper is simpler and more cost-effective.
2.	The use of SDS (Surface Distance Similarity) loss to align the reconstructed object mesh with the multi-view fused point cloud enhances the robustness of object pose estimation.
3.	The method incorporates hand keypoint estimation to refine the accuracy of hand pose prediction.
4.	A 3D-to-2D projection strategy is employed to further improve 2D hand keypoint detection.

---

> ### Author Rebuttal · Authors · 2025-07-30
>
> > **Q1: In the current loss design, hand and object are still optimised largely separately; the relation between them appears weak. Stronger hand–object interaction constraints (e.g., contact optimisation) could be explored.**
>
> **A1:**
> We agree that our current optimization framework primarily focuses on independently refining hand and object poses, with weak coupling via the joint optimization stage. And we fully agree that incorporating explicit contact constraints (e.g., minimizing mesh distance or enforcing grasp stability) would strengthen the physical plausibility of annotations.
>
> We are actively exploring the integration of contact-aware loss functions into our pipeline and plan to include these enhancements in future updates of the HO-Cap dataset and annotation framework.
>
> > Q2: **The dataset contains only 64 videos, roughly one-tenth of OakInk2(Mocap). Since the annotation is advertised as lightweight, why not collect more data?**
>
> **A2:**
> We acknowledge that the current HO-Cap release contains 64 annotated video sequences, which is smaller in scale compared to OakInk2. However, our goal with this initial release is to demonstrate the feasibility, quality, and scalability of our lightweight, markerless annotation pipeline. And we would keep expanding the dataset with additional subjects, object categories, tasks, and sequences in future release updates.
>
> > **Q3: In the experiments, the authors treat hand-pose estimation and object-pose estimation as two separate tasks. However, given that the paper’s title emphasizes hand-object interaction, it lacks interaction-specific metrics—such as evaluating how well the hand and object surface meshes make contact within the dataset. Aside from briefly mentioning joint hand-object pose estimation, the method largely appears to tackle hand and object estimation independently.**
>
> **A3:**
> Thanks for the insightful comment. Following A1, we agree that our current evaluation primarily focuses on hand and object pose estimation independently, without explicitly measuring interaction quality. We will enhance the evaluation by including interaction-specific metrics, such as contact quality and grasp stability, in the final version of the paper.
>
> As a first step, we evaluated two interaction-specific metrics from the ObMan[1] paper: **penetration depth** and **intersection volume**. As shown in the table below, our current SDF-based optimization pipeline implicitly reduces mesh penetration and achieves the promised contact qyality:
>
> | Stage                 | Penetration Depth ($mm$) | Intersection Volume ($cm^3$) |
> | --------------------- | ------------------------ | ---------------------------- |
> | SDF-Refined Poses     | $6.20 \pm 3.29$          | $3.88 \pm 2.90$              |
> | Jointly-Refined Poses | **$6.01 \pm 3.17$**      | **$3.45 \pm 2.50$**          |
>
> > **Q4: Please add a comparison with the HOT3D dataset.**
>
> **A4:**
> Thank you for pointing out HOT3D, which we missed in the original submission. We will include a comparison and discussion in the related work section.
>
> | Dataset | Year | Modality       | #Seq. | #Frames | #Subj. | #Obj. | #Views | Real Image | Markerless | Bimanual | Object Reconst. | Task       | Label |
> | ------- | ---- | -------------- | ----- | ------- | ------ | ----- | ------ | ---------- | ---------- | -------- | --------------- | ---------- | ----- |
> | HOT3D   | 2024 | RGB-Monochrome | 424   | 3.4M    | 19     | 33    | 2+ego  | ✓          | ✗          | ✓        | ✓               | Multi-task | MoCap |
>
> > **Additional Feedback (Summary): Does the limited activity diversity reflect a weakness of the estimation method, and is the interaction aspect underrepresented despite being emphasized in the title?**
>
> **ADD1:**
> We appreciate the reviewer’s comments. HO-Cap’s primary goal is to demonstrate a scalable, markerless annotation pipeline using foundation models and multi-view RGB-D data for hand-object interaction.
>
> The initial release focuses on three core activity types, chosen to capture dense contact, multi-object manipulation, and occlusion-rich scenarios.
>
> The limitations of our pipeline are discussed in Section 7, but current activity scope does not reflect a limitation of the estimation method. Our annotation pipeline is generalizable to a broader range of tasks, and we are actively expanding HO-Cap with more activities, participants, and interaction dynamics in future releases.
>
> Regarding the interaction aspect: the title emphasizes our pipeline focuses on realistic hand-object interaction capture.
>
> [1] "Learning joint reconstruction of hands and manipulated objects", CVPR 2019.

---

> > ### Comment · Reviewer_xudw · 2025-08-04
> >
> > Thank you for the thorough response.
> >
> > The authors clarified the reasons for the current dataset’s limited size. I encourage the authors to scale up the dataset in future iterations. These responses address my concern.
> >
> > I will raise my overall rating.

---

> > > ### Author Response · Authors · 2025-08-04
> > > **Thank You!**
> > >
> > > Dear Reviewer xudw,
> > >
> > > Thank you for your thoughtful feedback on our rebuttal. We’re glad to hear that our response addressed your concerns. We will follow your suggestion to include the additional results in the final version and will continue to expand the HO-Cap dataset in future updates. We sincerely appreciate your support and encouragement.
> > >
> > > Sincerely,
> > > The Authors

---

### Official Review · Reviewer_7WLg · 2025-07-02

**Rating:** 5
**Confidence:** 4

**Summary:**

The submission includes details of a new dataset for hand-object interaction, covering humans performing tasks like grasping and multi-object interaction.  The submission includes details about a new capture system as well a semi-automatic annotation pipeline that can get annotations for 3D shapes and poses for hands and objects from multi-view RGB-D videos using pretrained vision models and SDF optimization, without domain specific object trackers.
Compared to mocaps, the proposed capture system captures marker less data without needing calibration between markers and cameras. The semi-automated annotation pipeline is proposed as a far more efficient alternative to manual labelling. The pipeline consists of multiple high quality The capture system includes 8 Intel Realsense D455 and one Azure Kinect that are calibrated. Ego-centric view is taken from HoloLens.

**Dataset Code Accessibility:**

Partly

**Dataset Code Comments:**

I was able to access the dataset files on huggingface but the quick copy-paste python snippet throws errors indicating that the path does not exist on hf. If the snippet is correct, can you provide more detailed instructions on loading the dataset?

**Ethical Considerations:**

No, there are no or only very minor ethics concerns

**Final Justification:**

Thanks to the authors for addressing my concerns and questions. This paper proposes a Hand-object interaction dataset which is richly annotated and provides a semi-automatic pipeline to ease the process of scaling the dataset up. The authors also addressed the issues about accessing the dataset via the code which is very helpful.
I would like to stand by my original positive rating of "5: Accept " since I believe this is a strong and impactful contribution to the community. The dataset still has some limitations that could be addressed in future revisions, and the proposed methods are not groundbreaking enough for a rating of 6, in my opinion.

**Limitations Weaknesses:**

1. Seems like the single-hand samples in the dataset is skewed towards right hands.
2. Hand pose estimators should’ve been fine-tuned on the new dataset to quantify an improvement in pose estimation results on the test set and in-the-wild Hand-object data. This would help to show the efficacy of this dataset for hand pose estimation task in scenario's of object manipulation.
3. Limitations about the annotation method as the authors described in Section 7. eg. this dataset lacks high quality annotation for texture-less objects etc. but the authors have already acknowledged and explained the cause of this limitation.
4. While the dataset seems to be very useful, it only includes 9 participants, so having such few identities could lead to worse generalizability in use-cases of this dataset.

**Strengths Contributions:**

1. The submission includes a hand object dataset and a fairly scalable marker less way of collecting a new data. Collecting data like this is challenging and expensive, so this is a useful contribution to the community.
2. Use of semi-automated annotation pipeline makes the data annotation much more scalable than manual annotation.
3. The proposed method does not require domain specific pose tracker as past datasets like H2O which trains an object pose tracker based on 3d reconstructions of each object.
4. The annotation pipeline contains rich ground truths for 3D hand pose shapes, 3D object pose and shape, segmentations and key points.
5. Performance of baseline hand pose estimation on this dataset, shows that this dataset is challenging for existing hand pose estimators due to shortage of data that includes hand-object occlusions.

---

> ### Author Rebuttal · Authors · 2025-07-30
>
> > **Q1: Seems like the single-hand samples in the dataset is skewed towards right hands.**
>
> **A1:**
> We acknowledge the reviewer’s observation regarding the imbalance toward right-hand sequences. Among the 9 participants, 2 are left-handed and were instructed to perform tasks according to their natural preference, which resulted in a natural skew toward right-hand usage.
>
> While this reflects real-world population statistics, we agree that including more left-handed participants would enhance hand-side diversity. We plan to address this in future extensions of HO-Cap by expanding the subject pool with a more balanced distribution.
>
> > **Q2: Hand pose estimators should’ve been fine-tuned on the new dataset to quantify an improvement in pose estimation results on the test set and in-the-wild Hand-object data. This would help to show the efficacy of this dataset for hand pose estimation task in scenario's of object manipulation.**
>
> **A2:**
> We thank the reviewer for this valuable suggestion. We agree that fine-tuning hand pose estimators on HO-Cap and evaluating them on the test set—as well as in-the-wild data—would better demonstrate the dataset’s value for improving hand pose estimation in manipulation scenarios.
>
> We are currently conducting experiments with fine-tuned models and will include the updated results and analysis in the final version and supplementary material.
>
> > **Q3: Limitations about the annotation method as the authors described in Section 7. eg. this dataset lacks high quality annotation for texture-less objects etc. but the authors have already acknowledged and explained the cause of this limitation.**
>
> **A3:**
> We appreciate the reviewer’s understanding of the annotation limitations.
>
> > **Q4: While the dataset seems to be very useful, it only includes 9 participants, so having such few identities could lead to worse generalizability in use-cases of this dataset.**
>
> **A4:**
> We acknowledge that the dataset currently includes a limited number of participants (9), which may affect generalizability in certain use cases. However, our pipeline is designed to be scalable, and we plan to expand the dataset with more participants in future updates. In the meantime, HO-Cap already includes 64 object categories and over 100K annotated frames across diverse tasks, scenes, and camera views, which helps counterbalance the identity limitation.
>
> > **Dataset Code Comments: dataset access and loading instructions.**
>
> Thanks for the comments! Due to HuggingFace Datasets’ naming rules and file size limitations, we are unable to upload the raw dataset directly. After downloading the .tar files, please extract and organize the data according to the following folder structure:
>
> - calibration.tar
>
> ```
> extrinsics.extrinsics_20231014.yaml ==> calibration/extrinsics/extrinsics_20231014.yaml
> intrinsics.037522251142.yaml ==> calibration/intrinsics/037522251142.yaml
> mano.subject_1.yaml ==> calibration/mano/subject_1.yaml
> ...
> ```
>
> - models.tar
>
> ```
> G01_1.cleaned_mesh_2000.obj ==> models/G01_1/cleaned_mesh_2000.obj
> G01_1.cleaned_mesh_10000.obj ==> models/G01_1/cleaned_mesh_10000.obj
> G01_1.textured_mesh.obj ==> models/G01_1/textured_mesh.obj
> G01_1.textured_mesh.png ==> models/G01_1/textured_mesh.png
> G01_1.textured_mesh.mtl ==> models/G01_1/textured_mesh.mtl
> ...
> ```
>
> - sequence_labels.tar
>
> ```
> subject_1-20231025_165502.meta.yaml ==> subject_1/20231025_165502/meta.yaml
> subject_1-20231025_165502.poses_m.npy ==> subject_1/20231025_165502/poses_m.npy
> subject_1-20231025_165502.poses_o.npy ==> subject_1/20231025_165502/poses_o.npy
> subject_1-20231025_165502.poses_pv.npy ==> subject_1/20231025_165502/poses_pv.npy
> ```
>
> - subject_1-20231025_165502.tar
>
> ```
> 037522251142-000000.color.jpg ==> subject_1/20231025_165502/037522251142/color_0000000.jpg
> 037522251142-000000.depth.png ==> subject_1/20231025_165502/037522251142/depth_000000.png
> 037522251142-000000.label.npy ==> subject_1/20231025_165502/037522251142/label_000000.npz
> ...
> ```
>
> Due to HuggingFace Datasets’ naming rules and file size limitations, we are unable to upload the raw dataset directly. After downloading the .tar files, please extract and organize the data according to the following folder structure:
>
> ```
> datasets/HO-Cap
>   ├── calibration
>   ├── models
>   ├── subject_1
>   │   ├── 20231025_165502
>   │   │   ├── 037522251142
>   │   │   │   ├── color_000000.jpg
>   │   │   │   ├── depth_000000.png
>   │   │   │   ├── label_000000.npz
>   │   │   │   └── ...
>   │   │   ├── 043422252387
>   │   │   ├── ...
>   │   │   ├── hololens_kv5h72
>   │   │   ├── meta.yaml
>   │   │   ├── poses_m.npy
>   │   │   ├── poses_o.npy
>   │   │   └── poses_pv.npy
>   │   ├── 20231025_165502
>   │   └── ...
>   ├── ...
>   └── subject_9
> ```
>
> We will update our code repository with detailed instructions for dataset access and loading, including the necessary scripts to handle the data structure and format.

---

> > ### Comment · Reviewer_7WLg · 2025-08-08
> >
> > > While this reflects real-world population statistics, we agree that including more left-handed participants would enhance hand-side diversity. We plan to address this in future extensions of HO-Cap by expanding the subject pool with a more balanced distribution.
> >
> > Thanks for acknowleding the limitation. I would encourage the authors to address this in future revisions of the dataset.
> >
> >
> > > ... We are currently conducting experiments with fine-tuned models and will include the updated results and analysis in the final version and supplementary material.
> >
> > This would be an important experiment and I'm glad the users are adding it to the final version and/or the supplemental material.
> >
> >
> > The other concerns have been addressed and clarified. I thank the authors for the clear response.
> >
> > I have no additional concerns. I will maintain my original rating.

---

> > > ### Comment · Area_Chair_5QXr · 2025-08-08
> > >
> > > Dear Reviewer 7WLg,
> > >
> > > Thank you again for reviewing the paper.  Please complete the Final Justification before Aug 13th.
> > >
> > > Best,
> > > AC

---

### Official Review · Reviewer_4Cyz · 2025-07-03

**Rating:** 4
**Confidence:** 4

**Summary:**

This paper introduces a novel capture system and dataset for joint 3D tracking of hand-object interaction. They develop a multi-view setup comprising eight calibrated RGB-D cameras and a HoloLens headset to provide both third-person and ego-centric views. Their core contribution is a semi-automatic annotation pipeline that reconstructs 3D object models and estimates poses through a multi-stage optimization process. This pipeline leverages large pre-trained models while minimizing manual intervention to just two-point initialization per object, achieving markerless 3D object reconstruction without costly scanners. The proposed SDF-based joint optimization further refines both hand and object poses while enforcing temporal consistency and physical plausibility. The authors created a new dataset for hand-object interaction research, covering diverse grasping and multi-object rearrangement tasks. The dataset establishes benchmarks with baseline results for object detection, hand pose estimation, and object pose estimation, providing valuable demonstrations for future research.

**Dataset Code Accessibility:**

Yes

**Dataset Code Comments:**

This paper provides complete data collection and annotation code, experimental code, and dataset.

**Ethical Considerations:**

No, there are no or only very minor ethics concerns

**Final Justification:**

This paper proposes a new dataset for hand-object interaction. By leveraging the capabilities of existing foundation models, it significantly improves the efficiency of  annotation. The author also explains the difference from existing datasets in the rebuttal. Therefore, my final score is borderline accept.

**Limitations Weaknesses:**

- The joint hand-object pose optimization yields marginal quantitative improvement (14.26->14.25 for left hand and 11.52->11.51 for right hand ) and lacks qualitative analysis (e.g. visualizations of reduced mesh interpenetration), undermining claims of enhanced physical realism.
- Despite permitting unrestricted interaction styles, ​​the absence of deliberate diversity-promoting mechanisms, coupled with a small cohort size (n=9), risks convergent manipulation patterns across subjects for identical objects, undermining dataset variability claims.​​
- The paper fails to discuss [GraspNet24] published in ECCV 2024, which present contextual similarities.

**Strengths Contributions:**

- The multi-camera capture system and object reconstruction framework significantly reduces data collection costs while maintaining high-quality annotations.
- The SDF-based optimization method yields hand and object pose annotations comparable to manual labeling, with quantitative validation confirming minimal error margins.

---

> ### Author Rebuttal · Authors · 2025-07-30
>
> > **Q1: The joint hand-object pose optimization yields marginal quantitative improvement (14.26->14.25 for left hand and 11.52->11.51 for right hand ) and lacks qualitative analysis (e.g. visualizations of reduced mesh interpenetration), undermining claims of enhanced physical realism.**
>
> **A1:**
> We thank the reviewer for this observation. To address the imporvement of physical plausibility:
>
> - We will provide **qualitative visualizations** in the supplementary material, including side-by-side comparisons of mesh interpenetration before and after joint optimization.
> - To show the **quantitate results**, we evaluated two interaction-specific metrics from the ObMan paper: **penetration depth** and **intersection volume**. As shown in the table below, our jointly-refined poses reduce both metrics compared to the SDF-refined baseline:
>
> | Stage                 | Penetration Depth ($mm$) | Intersection Volume ($cm^3$) |
> | --------------------- | ------------------------ | ---------------------------- |
> | SDF-Refined Poses     | $6.20 \pm 3.29$          | $3.88 \pm 2.90$              |
> | Jointly-Refined Poses | **$6.01 \pm 3.17$**      | **$3.45 \pm 2.50$**          |
>
>
> > **Q2: Despite permitting unrestricted interaction styles, ​​the absence of deliberate diversity-promoting mechanisms, coupled with a small cohort size (n=9), risks convergent manipulation patterns across subjects for identical objects, undermining dataset variability claims.​​**
>
> **A2:**
> We acknowledge the reviewer’s concern that manipulation strategies may converge for certain object categories (e.g., common tools). As our goal was to capture natural, unconstrained human behaviors, we did not explicitly enforce diversity during data collection.
> Nevertheless, HO-Cap includes multiple tasks, varied object shapes/sizes, and diverse initial object layouts and hand-object spatial relations across over 650K annotated frames.
> This helps promote behavioral variability despite the limited number of participants (n=9).
>
> Moreover, our pipeline is designed to be scalable, and we plan to expand the dataset with more participants and behavioral diversity in future updates.
>
> > **Q3: The paper fails to discuss [GraspNet24] published in ECCV 2024, which present contextual similarities.**
>
> **A3:**
> We thank the reviewer for pointing out **HOGraspNet** [1], which we missed in the original submission. We will include a detailed discussion and comparison in the revised related work section.
>
> | Dataset      | Year | Modality | #Seq. | #Frames | #Subj. | #Obj. | #Views | Real Image | Markerless | Bimanual | Object Reconst. | Task   | Labeling Method     |
> |--------------|------|----------|--------|---------|--------|--------|--------|-------------|-------------|-----------|------------------|--------|----------------------|
> | HOGraspNet | 2024 | RGB-D    | —      | 1.5M    | 30     | 99     | 4      | ✓           | ✗           | ✗         | ✗                | Grasp  | MoCap & Semi-auto   |
>
>
> [1] "Dense Hand-Object (HO) GraspNet with Full Grasping Taxonomy and Dynamics", ECCV2024.

---

> > ### Comment · Reviewer_4Cyz · 2025-08-06
> >
> > Thanks for the author's detailed response, which has addressed most of my concerns. Although I still argue that this dataset does not have an obvious advantage compared to some of the latest datasets (HOGraspNet), I think it is helpful for the field. I hope the author can further elaborate on the uniqueness of this dataset.

---

> > > ### Author Response · Authors · 2025-08-06
> > >
> > > We appreciate your feedback and the opportunity to clarify the uniqueness of HO-Cap dataset.
> > >
> > > Below, we summarize the key aspects that differentiate HO-Cap from other datasets like HOGraspNet:
> > >
> > > - **Joint Egocentric and Exocentric Views:** HO-Cap provides synchronized egocentric (HoloLens) and third-person multi-view RGB-D recordings. This setup enables research on cross-view learning, egocentric perception, and domain adaptation across perspectives. These capabilities are not supported by existing datasets such as DexYCB, HO-3D, OakInk, or HOGraspNet, which primarily focus on exocentric views.
> > > - **Markerless, Real-World Capture:** HO-Cap captures natural, real-world hand-object interactions without using reflective markers, thereby avoiding artifacts common in mocap-based pipelines. In contrast, datasets such as ARCTIC, OakInk, and HOGraspNet require markers or highly controlled settings.
> > > - **Rich Interaction Scenarios:** HO-Cap includes diverse manipulation scenarios such as multi-object rearrangement, tool usage, and bimanual interactions. This goes beyond the short, single-object interactions typical in DexYCB, HO-3D, OakInk, and HOGraspNet, offering a more realistic and challenging benchmark.
> > > - **Diverse Grasp Types:** Although task-driven, HO-Cap captures 28 out of the 33 grasp types defined by Feix et al. [1], significantly more than HO-3D (15) and DexYCB (14), and comparable to HOGraspNet. This highlights the dataset’s coverage of natural grasping behaviors in real-world tasks.
> > > - **Scalable and Cost-Efficient Annotation Pipeline:** HO-Cap provides 64 daily objects and 9 hand shapes, and supports rapid expansion to novel objects and unseen hand shapes without requiring pre-existing CAD models or scanning. This makes it ideal for few-shot and open-set learning tasks. The light hardware and low-labor semi-automatic pipeline encourages future community-driven dataset growth. In contrast, datasets like HOGraspNet rely on pre-scanned object shapes, expensive motion cameras and heavy setup, limiting their extensibility.
> > > - **Integration of Foundation and Neural Models:** HO-Cap is among the first datasets to integrate SAM2, FoundationPose, MediaPipe, and BundleSDF into a unified annotation pipeline, operating under model-free, occlusion-rich, real-world settings. It serves as a testbed for evaluating and extending foundation models within hand-object understanding pipelines, a growing trend in the community.
> > > - **Extended Applications:** Like the further possibilities addressed by HOGraspNet, HO-Cap also supports synthetic augmentation via texture and background variation. Beyond that, HO-Cap annotations can be replayed in simulators (e.g., IsaacGym, as shown in the supplementary video), enabling robotic agents to learn hand-object interaction skills. HO-Cap also includes both grasping and non-grasping interactions in multi-object scenes, making it more suitable for realistic robotic learning.
> > >
> > > In summary, HO-Cap is a realistic, scalable, and versatile platform for hand-object interaction research. Its combination of egocentric and exocentric views, diverse grasp types, rich real-world interactions, and integration of foundation models makes it both unique among existing datasets and a strong complement to HOGraspNet.
> > >
> > > [1] Feix, T., et al. "The grasp taxonomy of human grasp types". IEEE Transactions on Human-Machine Systems (2015).

---

### Note · Authors · 2025-08-16

We appreciate all reviewers’ thoughtful feedback and constructive suggestions. We have carefully considered each point raised and will address them in the revised version of our paper, including the following key updates:

- **Related Work Discussion:** We will include a more detailed comparison with recent datasets, including HOGraspNet [1] and HOT3D [2], in the revised Related Work section.
- **Evaluation of Fine-Tuned Hand Pose Estimators:** We will add quantitative results showing the performance of hand pose estimators fine-tuned on HO-Cap, to better demonstrate the dataset’s value for improving hand pose estimation.
- **Improved Physical Plausibility:** To address concerns about annotation quality, we will provide qualitative visualizations and report quantitative evaluations using interaction-specific metrics at each stage of our annotation pipeline.
- **Dataset Contributions:** We will include a quantitative analysis of the grasp types observed in HO-Cap based on the taxonomy defined by Feix et al. [3]. This highlights the dataset’s coverage of natural grasping behaviors across diverse task scenarios.
- **Future Dataset Expansion:** We will continue to expand the HO-Cap dataset in future updates by incorporating a more balanced distribution of left/right hand usage and enriching the diversity of interaction scenarios.

[1] "Dense Hand-Object (HO) GraspNet with Full Grasping Taxonomy and Dynamics", ECCV2024.\
[2] "Hot3d: Hand and object tracking in 3d from egocentric multi-view videos", CVPR2025.\
[3] Feix, T., et al. "The grasp taxonomy of human grasp types". IEEE Transactions on Human-Machine Systems (2015).

---

### Decision · Program_Chairs · 2025-09-18

**Decision:**

Accept (poster)

**Comment:**

The paper introduces HO-Cap, a novel and relatively low-cost system for capturing and annotating 3D hand–object interaction data. The setup includes eight calibrated RGB-D cameras plus an egocentric HoloLens, producing both third-person and first-person views. A semi-automatic annotation pipeline leverages large pre-trained models (e.g., SAM2 for segmentation, FoundationPose for tracking) along with SDF/SDS-based optimization, requiring only minimal manual initialization. This allows markerless 3D object reconstruction and joint hand–object pose estimation without CAD models or motion-capture systems. The resulting dataset (64 videos, 656K frames, 9 participants, 64 objects) covers diverse grasping and multi-object tasks, and provides baselines for object detection, hand pose estimation, and object pose estimation.

One of the major strengths of this paper is its efficiency. The capture hardware relies on off-the-shelf devices and does not require 3D scanning of target objects. The annotation pipeline leverages existing pretrained models to provide a semi-automatic, cost-effective process, requiring only the manual selection of two points per object in the first frame for initialization.

The main concerns from the initial review were its novelty, scale, and diversity. While the cost-effectiveness of the capture process is a valuable contribution, it remains unclear what unique scenarios or attributes HO-Cap provides beyond prior datasets. In addition, the dataset includes only 9 subjects and 64 videos, with an apparent imbalance toward right-hand sequences (7 right-handed, 2 left-handed).

In the author–reviewer discussion, the authors provided further explanations and experimental results that adequately addressed the reviewers’ concerns. All reviewers then recommended acceptance, and the AC would like to follow their recommendations.